# Loss of function mutations in GEMIN5 cause a neurodevelopmental disorder

Sukhleen Kour [1,52], Deepa S. Rajan[1,52], Tyler R. Fortuna[1], Eric N. Anderson[1], Caroline Ward[1], Youngha Lee[2], Sangmoon Lee[2], Yong Beom Shin[3], Jong-Hee Chae[4], Murim Choi [2,4], Karine Siquier[5], Vincent Cantagrel [5], Jeanne Amiel[6], Elliot S. Stolerman[7], Sarah S. Barnett[8], Margot A. Cousin [9], Diana Castro[10], Kimberly McDonald[11], Brian Kirmse[12], Andrea H. Nemeth[13,14], Dhivyaa Rajasundaram[15], A. Micheil Innes [16], Danielle Lynch[16], Patrick Frosk [17], Abigail Collins[18], Melissa Gibbons[18], Michele Yang[18], Isabelle Desguerre[19], Nathalie Boddaert[20], Cyril Gitiaux[21], Siri Lynne Rydning [22], Kaja K. Selmer[23], Roser Urreizti [24], Alberto Garcia-Oguiza[25], Andrés Nascimento Osorio[26], Edgard Verdura[27], Aurora Pujol [27,28], Hannah R. McCurry[29], John E. Landers[30], Sameer Agnihotri[31], E. Corina Andriescu[32], Shade B. Moody[32], Chanika Phornphutkul[33], Maria J. Guillen Sacoto[34], Amber Begtrup [34], Henry Houlden [35], Janbernd Kirschner[36], David Schorling[36], Sabine Rudnik-Schöneborn[37], Tim M. Strom[38], Steffen Leiz[39], Kali Juliette[40], Randal Richardson[40], Ying Yang[41], Yuehua Zhang[41], Minghui Wang[42], Jia Wang[43], Xiaodong Wang [43], Konrad Platzer[44], Sandra Donkervoort[45], Carsten G. Bönnemann[45], Matias Wagner [46], Mahmoud Y. Issa [47], Hasnaa M. Elbendary[47], Valentina Stanley[48], Reza Maroofian[35], Joseph G. Gleeson [48], Maha S. Zaki [47], Jan Senderek[49] & Udai Bhan Pandey [1,50,51✉]

GEMIN5, an RNA-binding protein is essential for assembly of the survival motor neuron (SMN) protein complex and facilitates the formation of small nuclear ribonucleoproteins (snRNPs), the building blocks of spliceosomes. Here, we have identified 30 affected individuals from 22 unrelated families presenting with developmental delay, hypotonia, and cerebellar ataxia harboring biallelic variants in the *GEMIN5* gene. Mutations in GEMIN5 perturb the subcellular distribution, stability, and expression of GEMIN5 protein and its interacting partners in patient iPSC-derived neurons, suggesting a potential loss-of-function mechanism. GEMIN5 mutations result in disruption of snRNP complex assembly formation in patient iPSC neurons. Furthermore, knock down of *rigor mortis*, the fly homolog of human *GEMIN5*, leads to developmental defects, motor dysfunction, and a reduced lifespan. Interestingly, we observed that GEMIN5 variants disrupt a distinct set of transcripts and pathways as compared to SMA patient neurons, suggesting different molecular pathomechanisms. These findings collectively provide evidence that pathogenic variants in *GEMIN5* perturb physiological functions and result in a neurodevelopmental delay and ataxia syndrome.

A full list of author affiliations appears at the end of the paper.

Perturbing the physiological functions of RNA-binding proteins (RBPs) can lead to motor neuron diseases such as amyotrophic lateral sclerosis, and spinal muscular atrophy (SMA) among others[1,2]. RBPs are critical for regulating multiple molecular functions including splicing, localization, translation, and mRNA stability[3–5]. RBPs exert those functions by forming large complexes with other proteins such as small nuclear ribonuclear proteins (snRNPs)[6–8]. The snRNPs, consisting of SMN, GEMIN (2–8), and Smith (Sm) core proteins, are an essential component of spliceosomes and helps remove introns from pre-mRNAs to generate mRNAs[9–11].

GEMIN5 is a multifunctional protein with the ability to interact with several different RNA and protein targets through different functional domains[12–15]. GEMIN5 is highly conserved across different species, and has been shown to localize in the nucleus as well as in the cytoplasm, suggesting important functions in both cellular compartments[16,17]. GEMIN5 physically binds to snRNA via a specific $AU_{5–6}$ sequence located within the highly conserved Sm site and is flanked by a short stem loop which assists in delivery to the SMN complex[10,14,15,18,19]. The specific snRNP code helps GEMIN5 in distinguishing these snRNAs from other forms of cellular RNAs[14,19–21].

Defects in RNA-mediated gene expression control are a hallmark of several human disorders[1]. GEMIN5 controls the expression of SMN by regulating translation of its mRNA[21]. SMN protein levels determine the mRNA-binding activity of GEMIN5, which in turn allows SMN to regulate its own expression. Loss of SMN protein causes SMA (MIM 253300), a fatal motor neuron disease, and the degree of snRNP assembly defects correlates with SMN protein levels[22–24] and SMA severity. However, the effects of disrupting snRNP complex dynamics in the pathogenesis of other disorders has not been studied. The interaction between GEMIN proteins and SMN suggests that variants in GEMIN5 could also result in neurological disorders.

Here we describe the clinical and molecular spectrum of variants in GEMIN5 among 30 patients presenting with developmental delay, hypotonia, motor dysfunction, and cerebellar atrophy, suggesting that GEMIN5 variants give rise to a distinct clinical phenotype. Pathogenic GEMIN5 variants significantly reduced the expression of snRNP components (SMN, Gemin2, Gemin4, and Gemin6) as compared to controls, suggesting a potential disruption in the snRNP complex as a whole. shRNA-mediated knockdown (KD) of endogenous GEMIN5 perturbed snRNP complex assembly. Importantly, knock down of rigor mortis, the fly homolog of human GEMIN5, leads to motor dysfunction and developmental delay similar to human patients. Using an RNA-sequencing approach, we identified transcriptomic changes caused by GEMIN5 variants in patient iPSC neurons. Taken together, our data establishes bi-allelic variants in GEMIN5 as a cause of a distinct neurological cerebellar ataxia syndrome, through altered snRNP complex assembly.

## Results

**Biallelic GEMIN5 variants cause motor predominant developmental delay and cerebellar atrophy**. We evaluated a 3-year-old female patient of Caucasian origin, born to non-consanguineous parents, presenting with developmental delay, central hypotonia, and ataxia at our neurogenetics clinic (Fig. 1a). Magnetic resonance imaging (MRI) of the brain showed diffuse cerebellar atrophy (Fig. 1d). Extensive metabolic and genetic testing was unrevealing; an ataxia multi-gene panel which included trinucleotide repeat analysis and mitochondrial genome sequencing was negative. Clinical whole exome sequencing (WES) analysis of the patient led to the identification of a c.3203T>C; p.(Leu1068-Pro) homozygous variant in the GEMIN5 gene (Fig. 1 and

Supplementary Table 1). This variant was confirmed by Sanger sequencing and familial segregation testing was consistent with recessive inheritance (Supplementary Fig. 1). The parents, as well as unaffected siblings, were heterozygous for the variant and had no obvious neurological symptoms. Subsequently, we identified 27 additional patients in 19 unrelated families with biallelic GEMIN5 variants (Fig. 1a, Supplementary Fig. 1, detailed clinical summary and Supplementary Table 1).

All patients showed motor predominant developmental delays and were diagnosed within the first 2 years of life. Patients 4, 5, and 6 (Family 3 and 4) presented with severe hypotonia at birth and were evaluated for SMA. These three patients passed away before 3 years of age. Most other patients had easily elicitable reflexes and did not fit the classical phenotype of SMA. While cognitive and speech delays were seen in most patients, the development delay was predominantly motor (details in Supplementary data 1 and Supplementary Table 1). No motor or cognitive regression was found in any of the patients. 23 of the 30 patients had central hypotonia, however, the appendicular tone was variable and included concomitant spasticity with brisk reflexes in 13 of the 30 patients. All ambulatory patients had a gait ataxia.16 of the 30 patients had an electromyography (EMG) and nerve conduction velocity (NCV) (Supplementary Table 2) where 10 of these suggestive off neuropathic as opposed to motor neuron disease. 15 of the patients had a static phenotype, with 6 patients experiencing a progressive phenotype. Data on the clinical progression of the remaining 9 patients was unavailable.

Furthermore, the brain MRI in all patients revealed cerebellar atrophy. Patients 4, 5, and 6 (Family 3 and Family 4) had cerebellar atrophy on brain MRI which was performed prior to the age of 6 months, suggesting the possibility of cerebellar hypoplasia. Patient 20 (Family 13), patient 24, and 25 (Family 17) had progressive cerebellar atrophy on repeat imaging. Patient 4 (Family 3), patient 17 (Family 11), patient 20 (family 13), and patients 24 and 25 (Family 17) had a progressive phenotype on clinical evaluation

All in all, we identified 30 variants in GEMIN5, with four of them to be presumed loss-of-function (family 10, 15, 17, and 20), along with 22 missense variants. All variants are evolutionary conserved residues across various species and are currently rare, or absent in gnomAD (Supplementary Table 3). The missense variants are predicted to be pathogenic and probably damaging in nature by various in-silico prediction tools such as Polyphen-2, PROVEAN, SNAP2, muPRO, PhD SNP, and SIFT (Fig. 1b, Supplementary Table 4). Eight of the GEMIN5 missense variants are located in conserved alpha helixes in the monomer–monomer interface (p.His1364Pro, p.His923Pro, p.Ile988Phe, p. Ser1000-Pro, p. Ala1007Thr, p. Asp1019Glu, p.Leu1367Pro, and p. Leu1119Ser), whereas six missense variants (p. Ser73Pro, p. His162Arg, p.Asp210Tyr, Val611Met, p.Gly683Asp, and p. Asp704Glu) are located in the WD40 domain, and five variants (p.Tyr1282His, p.Tyr1286Cys, p. Tyr1286Asn, p. His1264Pro, and p. Leu1367Pro) are located in the RNA-binding site 1 (RBS1) (Supplementary Fig. 3). Six of the variants involve a proline substitution (p. Ser73Pro, p.His923Pro, p.Ser1000Pro, p.Leu1068-Pro, p.His1364Pro, and p.Leu1367Pro), an amino acid which is well-known for disrupting alpha helix secondary structure (Supplementary Fig. 3). Overall, these findings suggest that these highly conserved variants in GEMIN5 might perturb GEMIN5 structure and function(s), resulting in deleterious neurological symptoms.

**Pathogenic variants cause loss of GEMIN5 and snRNP complex proteins expression**. To understand the consequences of biallelic variants in GEMIN5, we reprogrammed peripheral blood

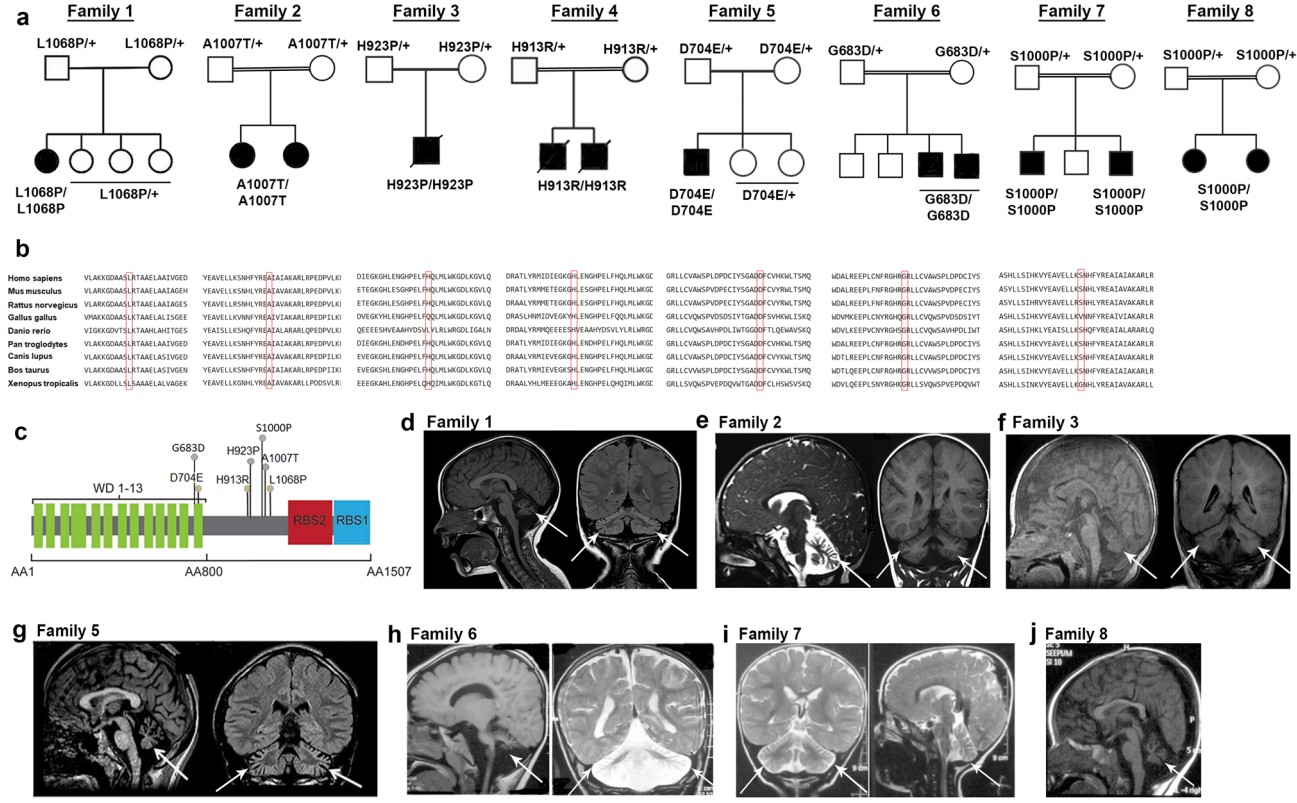

**Fig. 1 Variants in *GEMIN5* cause developmental delay, hypotonia, motor dysfunction, and cerebellar atrophy. a** Pedigree of the patients with homozygous variants in *GEMIN5*. Affected individuals who underwent clinical examinations are represented by arrowhead. **b** Multiple sequence alignment showing conservation of amino acid residues in GEMIN5 (red rectangle) across species. Polyphen-2 analysis predicted probably damaging effect of variants on GEMIN5 structure and function. **c** Schematic showing functional domains and position of homozygous variants in GEMIN5 protein. **d–j** MRI scans showing characteristic cerebellar atrophy (white arrow) in patients carrying different bi-allelic variants in *GEMIN5* (+ represents wild type allele of GEMIN5).

mononuclear cells (PBMCs) from the Leu1068Pro/Leu1068Pro patient and an unaffected parent carrying Leu1068Pro/+ into induced pluripotent stem cell lines (iPSC) (Supplementary Fig. 3a and b). Since the His913Arg patients were not alive, we used CRISPR/Cas9 to engineer the p.His913Arg heterozygote (referred herewith as control) and homozygous variants in a healthy control iPSC line. After doing extensive quality control testing of both iPSC lines, including sequencing and karyotyping analysis, we differentiated them into the neuronal cells (Supplementary Figs. 3c and 4). Two independent isogenic iPSC clones with homozygous p.His913Arg variant, His913Arg[A6], and His913Arg[A11] were used for the study.

GEMIN5 is predominantly a cytoplasmic protein with sparse nuclear localization under physiological conditions[16]. We asked if variants in *GEMIN5* perturb its subcellular expression pattern and localization in patient-derived iPSC neurons. By immunofluorescence (IF), we found a drastic decrease in the cytoplasmic distribution of GEMIN5 in the homozygous neuronal cells, p. His913Arg and p.Leu1068Pro, while neurons expressing heterozygous variants showed a normal physiological nuclear-cytoplasmic distribution of GEMIN5 (Fig. 2a–e). In contrast to His913Arg, homozygous Leu1068Pro neurons showed scattered punctate expression of GEMIN5 in the cytoplasm and these punctate structures do not co-localize with anti-GW182 (p-bodies marker) (Fig. 2a and Supplementary Fig. 5). No aberrant changes were seen in GEMIN5 nuclear levels between the homozygous and control groups (Fig. 2b and d). Since GEMIN5 is a critical component of the SMN complex involved in snRNP spliceosomal assembly[25], we next examined if mislocalization of mutant

GEMIN5 has any impact on the sub-cellular distribution pattern of other snRNP complex proteins such as SMN, GEMIN2, GEMIN4, and GEMIN6. We observed that GEMIN2 showed a similar distribution pattern of GEMIN5 in homozygous His913Arg and Leu1068Pro neurons (Fig. 2f). GEMIN2 levels showed a significant reduction in the cytoplasm with unaltered nuclear levels in homozygous His913Arg and Leu1068Pro neurons compared to heterozygous controls (Fig. 2g–j). On the other hand, SMN, GEMIN4, and GEMIN6 showed no obvious alterations in their distribution pattern between patient and control neurons (Supplementary Fig. 6).

GEMIN5 has previously been shown to be involved in global translational processes[26–28]. We investigated whether the GEMIN5 variants had any effect on the expression levels of GEMIN5 as well as its interacting partners of the SMN complex[16,25]. We found that the levels of GEMIN5 were drastically reduced by ~70–80% in Leu1068Pro and His913Arg patient neurons as compared to controls (Fig. 3a–d). Surprisingly, we also observed a significant reduction in the protein levels of GEMIN4, GEMIN3, GEMIN2, GEMIN6, SMN, and U1A in Leu1068Pro (Fig. 3a and b) and His913Arg (Fig. 3c, e–j) patient neurons as compared to controls. Consequently, to examine the possible underlying mechanisms responsible for the reduced intracellular levels of GEMIN5, we compared GEMIN5's protein stability between the Leu1068Pro patient and control neurons (Fig. 3k, l). We performed Western blot (WB) on the protein lysates harvested after 0, 2, 4, 8, 12, and 24 h of cycloheximide (CHX) treatment. WB analysis revealed an initial build-up of GEMIN5 for 4 h followed by a gradual drop off in control

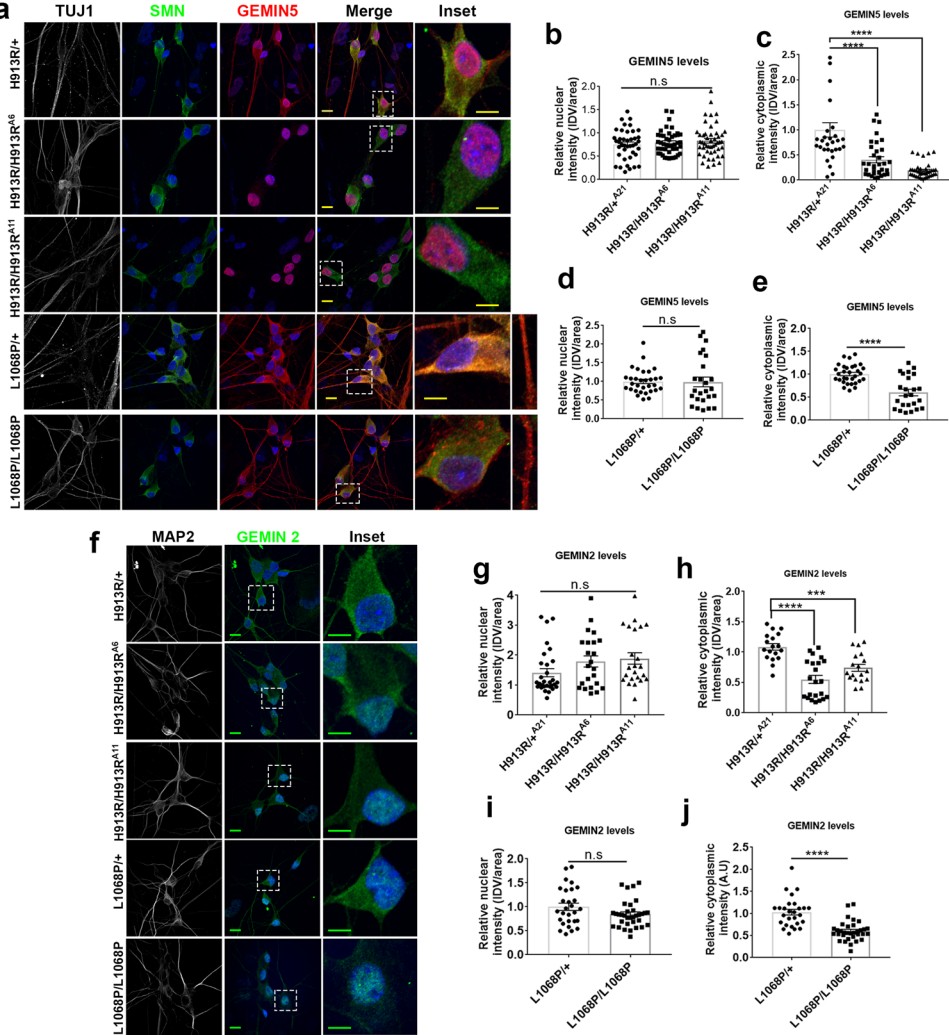

**Fig. 2 Differential subcellular expression of SMN assembly proteins in iPSC-derived neuronal cells carrying His913Arg and Leu1068Pro GEMIN5 variants. a** Representative Immunofluorescence (IF) images showing the sub-cellular mislocalization of GEMIN5 in IPSCs derived neuronal cells carrying p. His913Arg and p. Leu1068Pro hetero- and homozygous variants. H913R/H913R$^{A6}$ and H913R/H913R$^{A11}$ were the two isogenic iPSC clones for H913R homozygous variant used in the study (scale bar = 10 μm). **b, c** Quantitative analysis displaying the changes in nuclear (**b**) cytoplasmic (**c**) distribution pattern of GEMIN5 as in (**a**), measured as integrated density values (IDV) in H913R hetero and homozygous neuronal cells (one-way ANOVA- Bonferroni test, $n = 30$–40). **d, e** Quantitative analysis showing the nuclear (**d**) cytoplasmic (**e**) distribution of GEMIN5 as in (**a**) in L1068P hetero and homozygous neuronal cells (two tailed Mann–Whitney U test, $n = 25$–30). **f** Representative IF images showing the differential subcellular pattern of GEMIN2 in IPSCs-derived H913R and L1068P neuronal cells (scale bar = 10 μm). **g–j** Quantitative bar plot representing the integrated density values of GEMIN2 in the nucleus (**g, i**) and cytoplasm (**h, j**) of H913R (**g, h**) and L1068P (**i, j**) GEMIN5 neuronal cell soma (two tailed Mann–Whitney U test, $n \geq 25$). The data are represented as mean ± SEM. P values (****<0.0001, ***<0.001, **<0.01) are of unpaired Student's t test. Source data are provided as a Source Data file.

neurons, whereas we observed an initial reduction in GEMIN5 levels after 2 h of CHX treatment in Leu1068Pro patient neurons (Fig. 3k–m). Likewise, SMN protein levels showed a steady reduction after 2 h of CHX treatment in homozygous Leu1068Pro as compared to heterozygous neurons (Fig. 3n). No obvious changes were seen in GEMIN4 protein stability (Fig. 3o). Additionally, to address any possible link between reduced GEMIN5 protein levels and its stability with the degradation pattern, we examined the ubiquitination profile of the His913Arg homozygous and control neurons by IF. We found a robust increase in ubiquitinylated puncta in the cytoplasm and axons of homozygous His913Arg neurons as compared to heterozygotes (Supplementary Fig. 7). We further explored if the difference in GEMIN5 protein levels is due to differential expression and stabilities in their corresponding mRNAs. We performed qPCR to determine the basal expression of GEMIN5 mRNAs and found no significant difference in transcript levels between homozygous

and heterozygous His913Arg and Leu1068Pro neurons (Fig. 3p and q). To determine mRNA stability, we treated Leu1068Pro patient neurons with the global transcriptional inhibitor actinomycin D (ActD) for 0,1, 2, 4, 6, and 8 h and performed qPCR on the corresponding total RNAs (Fig. 3r). We found that GEMIN5 mRNAs are significantly less stable in Leu1068Pro homozygous neurons with the half-life ($t_{1/2}$) of 1.872 in contrast to $t_{1/2}$ of 2.559 in heterozygotes. The data suggests that the differential reduction of GEMIN5 in homozygous variants is due to difference in its mRNA and protein stability rather than transcriptional dysregulation.

**Knocking down endogenous GEMIN5 disrupts snRNP complex proteins and causes developmental delay and motor dysfunction in vivo.** Since homozygous Leu1068Pro and His913Arg variants led to a robust decrease in GEMIN5 and corresponding

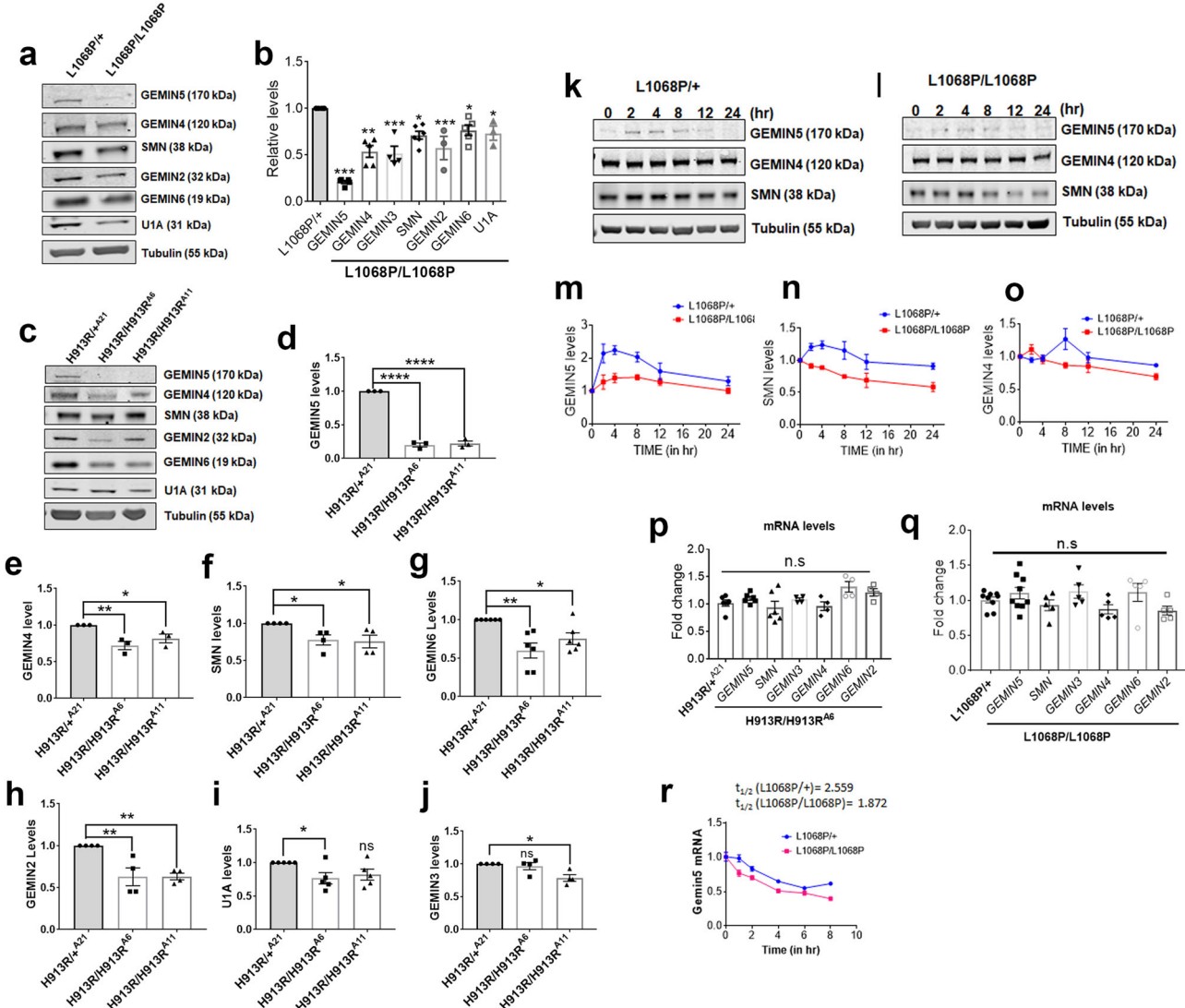

**Fig. 3 GEMIN5 variants reduce the levels of GEMIN5 and SMN assembly proteins. a** Western blot (WB) analysis of total protein extract from IPSC-derived neuronal cells carrying mono-allelic and bi-allelic L1068P *GEMIN5* variants depicting the levels of GEMIN5, GEMIN4, GEMIN6, and GEMIN2, SMN, and U1A, respectively**. b** Quantitative bar plots showing the reduced levels of GEMIN5 and SMN assembly proteins in L1068P homozygous neurons compared to heterozygous controls (two-tailed unpaired *t* test, n = 4). **c** Representative WB depicting the expression levels of GEMIN5 and SMN complex proteins in heterozygous and homozygous H913R neurons **d–j**, Quantitative comparison of expression levels of GEMIN5 (**d**), GEMIN4 (**e**), SMN (**f**), GEMIN6 (**g**), GEMIN2 (**h**), U1A (**i**), and GEMIN3 (**j**) between H913R$^{-/-}$ and H913R$^{+/-}$ neuronal cells as in (**d**) (one-way ANOVA-Bonferroni test, n = 4). **k, l** Representative WB showing the protein levels of GEMIN5, GEMIN4, and SMN in L1068P heterozygous (**k**) and homozygous (**l**) neurons after 0, 2, 4, 8, 12, and 24 h of cycloheximide (CHX) treatment. Tubulin was used as normalization control. **m–o** Quantitative analysis of the rate of degradation of GEMIN5 (**m**), SMN (**n**), and GEMIN4 (**o**) after CHX treatment showed the increased rate of depletion of GEMIN5 and SMN in homozygous L1068P neurons as compared to heterozygous controls (nonlinear regression-one phase decay, n = 4). **p, q** Expression analysis of GEMIN5, SMN, and other GEM-proteins by qPCR in L1068P (**p**) and H913R (**q**) neurons. The transcript levels of GEMIN5, GEMIN4, GEMIN3, GEMIN6, GEMIN2, and SMN showed no significant changes among heterozygous and homozygous GEMIN5, L1068P, and H193R variants (two tailed Mann–Whitney *U* test, n = 6). **r** Quantitative PCR showing the relative stability of GEMIN5 mRNA between hetero and homozygous L1068P neurons by using total RNA isolated at 0, 1, 2, 4, 6, and 8 h after actinomycin D treatment (nonlinear regression-one phase decay, n = 4). The data represent mean ± SEM. *P* values (****<0.0001, ***<0.001, **<0.01). Source data are provided as a Source Data file.

SMN complex proteins levels in differentiated neurons, we asked if the shRNA KD of GEMIN5 protein reciprocates the same effect as observed in the patient iPSC neurons. We transfected HEK293T cells with different shRNA constructs against GEMIN5 and measured the protein levels by WB (Supplementary Fig. 8). However, in order to get the robust KD of up to ~60–70%, similar to what was seen in homozygous patient iPSC neurons, we co-transfected HEK293T cells with two different combinations of shRNAs with the highest KD efficiency (shRNA B with shRNA 5

and 4) and evaluated the levels of SMN complex proteins by WB (Fig. 4a, Supplementary Fig. 8c and d). We observed that the effect of decreased GEMIN5 on members of the SMN complex is dosage-dependent, and significantly alleviated levels of SMN, GEMIN4, GEMIN3, GEMIN6, GEMIN2, and SmB1/B2 proteins only when GEMIN5 levels were reduced to below ~65% (Fig. 4b–l, Supplementary Fig. 8e–h). We also did reciprocal studies in HEK cells where we overexpressed different concentration of GEMIN5 to determine its subsequent effects on

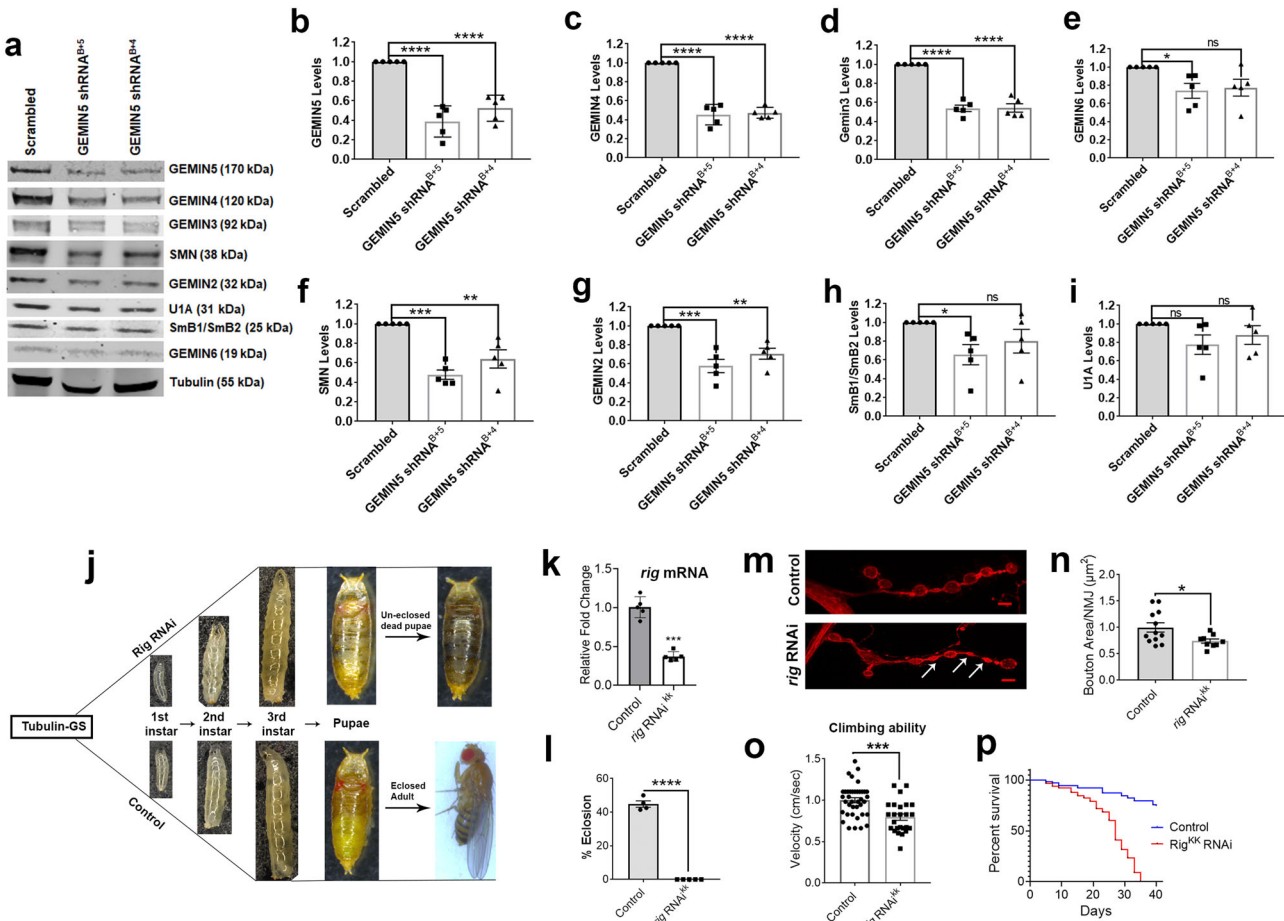

**Fig. 4 Loss of GEMIN5 leads to decrease levels of snRNP complex proteins and, developmental defects and motor dysfunction in *Drosophila*. a** Representative WB showing the effect of shRNA-mediated knockdown of GEMIN5 on the levels of GEMIN4, GEMIN3, GEMIN2, GEMIN6, SMN, SmB1/B2, and U1A as compared to scrambled control. GEMIN5 shRNA B was used in combination with GEMIN5 shRNA 5 and 4 to obtain the maximum knockdown efficiency. α tubulin was used as internal control (*n* = 5). **b–I** Quantitative analysis showed a significant decrease in the levels of GEMIN4 (**c**), GEMIN3 (**d**), GEMIN6 (**e**), SMN (**f**), GEMIN2 (**g**), and SmB1/B2 (**h**) upon ~65% knockdown of GEMIN5 (**b**). No significant change was found in U1A protein levels (**i**) (one-way ANOVA-Bonferroni test, *n* = 5). **j** Flow diagram comparing different developmental stages of flies between *rigor mortis* RNAi and W1118 control flies. RNAi-mediated knockdown of *rigor mortis*, as determined by qPCR in (**k**), resulted in pupal lethality (**j**) and eclosion defects (**l**) as measured by percentage eclosed adult homozygous flies (two tailed unpaired *t* test, *n* = 5). The RNAi transgene under inducible tubulin-UAS gal4 system was expressed by growing the larvae on 1 mM RU486 drug food. **m** Representative IF images of neuromuscular junction (NMJ) marked with HRP (pre-synaptic marker) in the larval segment expressing *rig mortis* RNAi compared to control (scale bar = 10 μm). **n** Quantitative comparison of the bouton size measured as area per NMJ between the rig mortis RNAi expressing larvae and control (two-tailed Mann–Whitney *U* test, *n* = 12). **o** Bar graph representing rapid iterative negative geotaxis (RING) assay, calculated as climbing speed of a fly per second, showed significant defects in the climbing velocities of flies with RNAi-mediated GEMIN5 KD as compared to controls. The effect was apparent when the transgene was expressed for 20 days on 20 mM RU486 drug food under the control of tubulin-GS driver (two-tailed Mann–Whitney *U* test, *n* = 25–39). **p** Kaplan–Meier survival plot showing the effect of the loss of endogenous *rig mortis* on the life span of flies. The flies were grown on 20 mM RU486 drug food to express the *rig mortis* RNAi transgene and monitored every day for the span of 45 days (log-rank (Mantel–Cox) test, *n* = 80). The data represent mean ± SEM. *P* values (****<0.0001, ***<0.001, **<0.01). Source data are provided as a Source Data file.

SMN complex proteins. Apart from GEMIN4, we observed no significant changes in the levels of SMN, U1A, SmB1/B2, and other GEM proteins (Supplementary Fig. 9).

In addition, we investigated the possible consequences of the loss of GEMIN5 in an in vivo *Drosophila* model. As the clinical manifestations related to *GEMIN5* variants occur at very early stages in humans, we asked if the loss of *rigor mortis*, a fly orthologue of human *GEMIN5*, by RNAi-mediated KD has any impact on the development of flies. We expressed RNAi transgene against *rigor mortis* in flies by using the inducible tubulin-GAL4/upstream activation sequence (UAS) system and monitored the development of flies from egg to adults on 1 mM RU486 drug food (Fig. 4j). We found complete pupal lethality in

the *rigor mortis* RNAi-expressing flies as compared to EGFP-controls (Fig. 4j and l), suggesting severe late-developmental defects with 60% loss of *rigor mortis* as validated by qPCR (Fig. 4k). Since the patients with GEMIN5 variants showed hypotonia and motor delay, we asked if neuronal KD of rigor mortis could cause motor function and neuromuscular junction (NMJ) defects. We first stained control and *rig mortis* KD animals with the pre-synaptic marker, horse radish peroxidase (HRP), to assess the NMJs (Fig. 4m). We found a significant reduction in the bouton size of larvae with rig mortis KD compared to the EGFP-controls (Fig. 4m, n). To examine further motor function defects, we performed rapid iterative negative geotaxis (RING) assay on neuronally expressing rigor mortis RNAi lines (Fig. 4m).

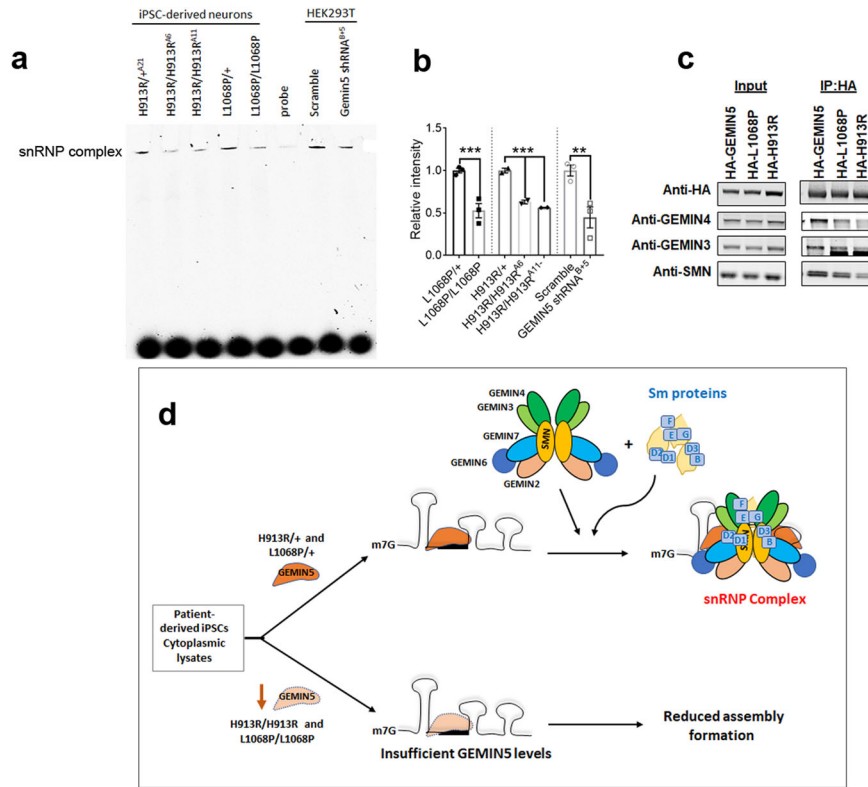

**Fig. 5 Biallelic variants in *GEMIN5* disrupts SMN assembly formation in vitro. a** Representative gel image showing the in vitro snRNP assembly formation by using 3′ Cy3-biotin-labeled U1 snRNA and the cytoplasmic extract from heterozygous and homozygous L1068P and H913R neurons as well as from the HEK293T cells transfected with GEMIN5 shRNA. **b** Quantitative analysis of the in vitro SMN complex formation as given in (**a**) (one-way ANOVA-Bonferroni test, $n = 3$). **c** Immunoprecipitation blot showing the reduced interaction of HA-tagged Leu1068Pro and His913Arg variants with GEMIN4, GEMIN3, and SMN as compared to HA-GEMIN5 WT protein in HEK cells. **d** Diagrammatic representation showing the possible mode of disruption in snRNP complex formation due to loss of GEMIN5 in L1068P and H913R variants. *P*-values ***<0.01, **<0.05. Source data are provided as a Source Data file.

We found that rigor mortis KD significantly reduced the climbing ability of adult flies compared to control animals (Fig. 4o). Three patients with biallelic *GEMIN5* variants showed early lethality and we identified loss of GEMIN5 protein in the homozygous patient-derived IPSCs neurons, so we investigated if the loss of GEMIN5 protein effects the life span of adult flies. We monitored flies expressing *rig mortis* KD ($n = 103$) over the span of 45 days and found 100% mortality in *rig mortis* KD flies after 33 days, as compared to 19% in w*1118* controls (Fig. 4p). Overall, we assessed that the loss of *rigor mortis* in vivo leads to premature lethality, motor dysfunctions, and reduced life span, which replicates the neurological symptoms found in GEMIN5 patients.

**GEMIN5 variants perturb snRNP complex formation in vitro.** GEMIN5 is an snRNA-binding protein that is essential for the spliceosomal snRNPs biogenesis[16,18]. To determine if the pathogenic GEMIN5 variants effected the assembly of core sm proteins in the SMN–snRNA complex, we decided to reconstitute the snRNP assembly in vitro by using in vitro transcribed 3′Cy3-biotinylated-U1snRNA and cytoplasmic extracts from Leu1068-Pro and His913Arg differentiated neurons. To examine the impact of loss of GEMIN5 on the assembly formation, we also used extract from HEK293T cells transfected with or without GEMIN5 shRNA. By native-PAGE, we found a distinct band representative of SMN-Sm assembly formation in control iPSC neurons and HEK293T control extracts. However, the assembly was drastically reduced in extracts from homozygous Leu1068Pro and His913Arg neurons as well as in HEK293T with GEMIN5 shRNA (Fig. 5a, b), suggesting that loss of GEMIN5 in Leu1068Pro and His913Arg neurons leads to disruption of

snRNP assembly formation (Fig. 5d). During assembly formation, GEMIN5 interacts with GEMIN3 and GEMIN4 and delivers pre-snRNA to SMN–GEMIN2–Sm protein complex. To assess if the reduced SMN assembly formation is related to the interaction of GEMIN5 variants with other GEM proteins and SMN, we performed immunoprecipitation by using anti-HA beads to affinity purify HA-tagged GEMIN5 WT, Leu1068Pro, and His913Arg variants and their interacting proteins in HEK-293T cells. As shown in Fig. 5c, the His913Arg and Leu1068Pro mutation in GEMIN5 drastically reduced GEMIN5's interaction with SMN, GEMIN4, and GEMIN3 as compared to WT, which could be driving the reduced snRNP assembly.

**GEMIN5 patient neurons show a distinct and unique transcriptomic signature as compared to SMA patient neurons.** GEMIN5 and SMN are part of the same ribonuclear–protein complex, but mutations in either of these proteins result in two distinct clinical presentations. These observations prompted us to ask if these differences could be explained by examining alterations in the transcriptomic profile of mutant homozygous GEMIN5 and SMA patient neurons. We performed RNA-seq analysis in iPSC-derived differentiated neurons with biallelic GEMIN5 (GEMIN5[H913R]) and compared this dataset with a published dataset from SMA (SMN1[Ex7del]) patient iPSC neurons. By using this in silico approach, we identified differentially expressed transcripts (DEGs) using a *P*-value threshold of ≤0.01 adjusted for statistical significance, and a log fold change of ≥1.5. Our analysis showed a consequential number of downregulated genes in GEMIN5[H913R] compared to SMN1[Ex7del] patient neurons (Fig. 6a, b). By comparing the significant DEGs in SMN1[Ex7del]

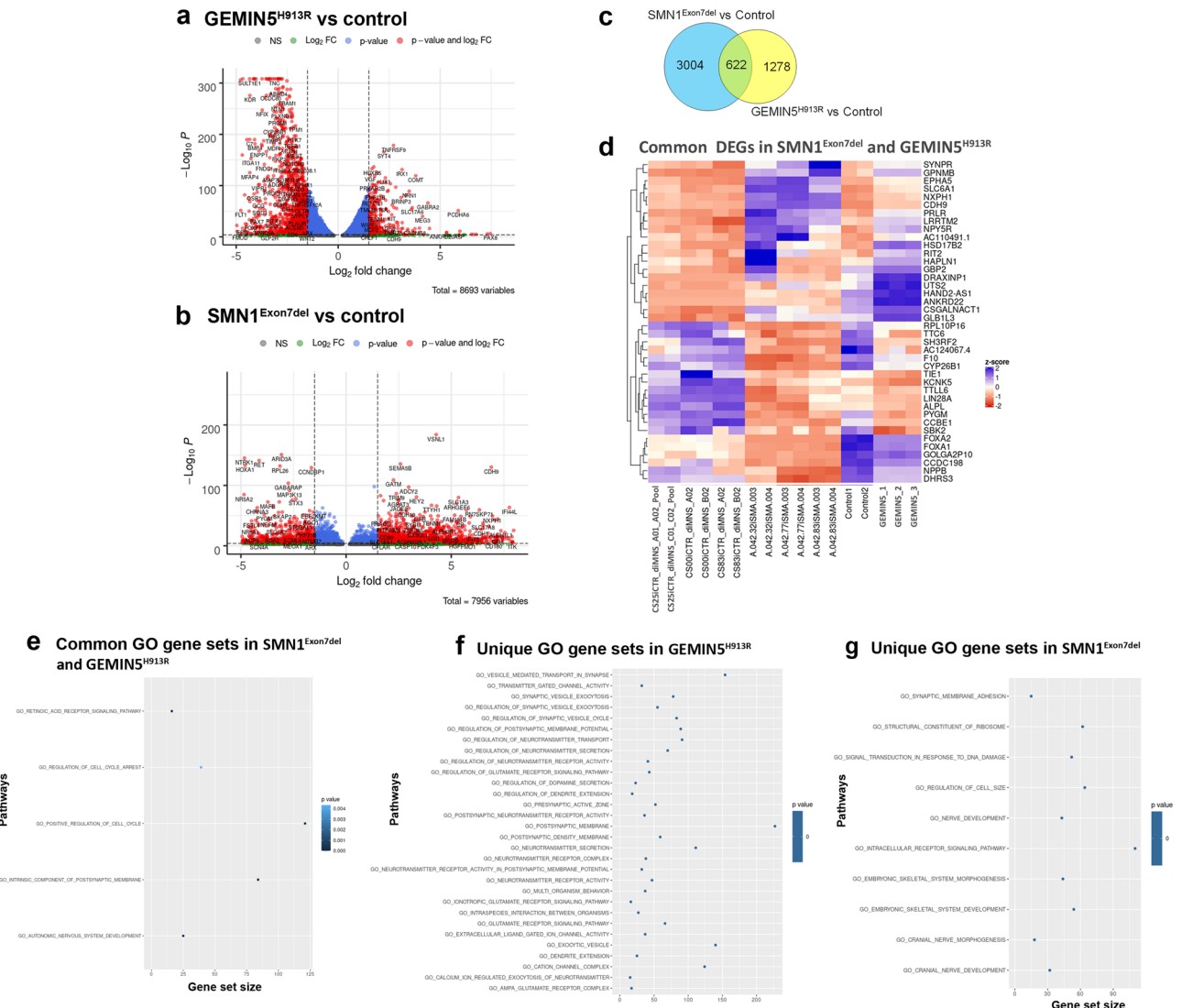

**Fig. 6 RNA-seq analysis of GEMIN5 patient iPSC neurons reveals a distinct and unique transcriptomic profile compared to SMA patient iPSC neurons.** **a**, **b** Volcano plot showing up and down-regulated genes in SMN$^{Exon7del}$ vs. control (**a**) and, GEMIN5$^{H913R}$ vs. control (**b**) selected by $p$-value < 0.01 & log2 (fold change) $\geq$ 1.5. The $x$-axes show log2 values of the fold changes in gene expression between the samples and $y$-axis shows $-$log10-transformed $p$-values. Significant genes were selected after Benjamini–Hochberg (BH) correction. **c** Venn diagram showing the number of genes that are shared between SMA and GEMIN5 and the genes which are exclusively to both. Only the DEGs with $\geq$1.5 log-fold DEGs were used for the comparison. **d** Heat map depicting the expression pattern of top 20 up and downregulated DEGs common to both SMN$^{Exon7del}$ and GEMIN5$^{H913R}$. Significant genes were selected by Wald test in DESeq2 and multiple test correction by BH. **e–g** Functional characterization of the genes with the MSigDB 'c5 Gene Ontology (GO), biological process ontology (BP) v6.0' gene sets in SMN$^{Exon7del}$ and GEMIN5$^{H913R}$ (**e**), unique to GEMIN5$^{H913R}$ (**f**) and SMN$^{Exon7del}$ (**g**). The size of the dot corresponds to the number of genes per term, and the color of the dot indicates the enrichment significance. Source data are provided as a Source Data file.

and GEMIN5$^{H913R}$ patient neurons as shown by the Venn diagram, we identified 1278 and 3004 transcripts unique to GEMIN5$^{H913R}$ and SMN1$^{Ex7del}$, respectively, whereas 622 transcripts are shared among these two disease conditions (Fig. 6c). Interestingly, heat map comparison with hierarchal clustering of the top 40 common DEGs in SMN1$^{Ex7del}$ and GEMIN5$^{H913R}$ showed a contrasting expression trend, suggesting that these two disease entities lead to distinct transcriptomic alterations (shown in red box in Fig. 6d). Specifically, we observed that a subset of transcripts upregulated in SMN1$^{Ex7del}$ showed an opposite downregulated trend compared to GEMIN5$^{H913R}$ iPSC neurons. DEGs exclusive to GEMIN5$^{H913R}$ are mostly involved in mRNA processing, brain development, neuronal transmission, and developmental processes, respectively (Supplementary Fig. 10). We validated GEMIN5-sequencing data by performing qPCR on

three highly upregulated (*SOX14*, *GBX2*, and *PDZRN4*) and three highly downregulated (*LRRC1*, *NXX2.1*, and *STX11*) genes (Supplementary Fig. 11).

Next, in order to mine the pathways which are either shared or unique to SMA and GEMIN5, we performed gene ontology (GO) and biological process ontology (BP)-enrichment analysis on the DEGs from both datasets and compared the top 30 identified pathways with adjusted $p$-value < 0.01 & log2(fold change) $\geq$ 1.5 between the two groups. Interestingly, we found that SMN1$^{Ex7del}$ and GEMIN5$^{H913R}$ shared only five notable pathways involved in the development of the autonomic nervous system, regulation of cell cycle, retinoic acid signaling, and postsynaptic membrane component (Fig. 6e). However, the majority of pathways altered in GEMIN5$^{H913R}$ are distinct from SMN1$^{Ex7del}$ which might explain why mutant GEMIN5 and SMA patients show different

clinical presentations (Fig. 6f and supplementary Fig. 10b). In addition, the pathways upregulated in GEMIN5[H913R] are associated with regulation of postsynaptic membrane potential, neurotransmitter secretion, transport, and signaling pathways (Fig. 6f), whereas the downregulated pathways were linked to regulation of developmental process, extracellular matrix organization, nuclear transport, and signal transduction (Supplementary Fig. 12). On the other hand, the pathways modulated in SMN1[Ex7del] were notably involved in nerve development and morphogenesis, intracellular receptor-signaling pathways, synaptic membrane adhesion, response to DNA damage, and regulation of ribosomal assembly (Fig. 6g). Overall, the transcriptomic comparison between SMN1[Ex7del] and GEMIN5[H913R] patient neurons suggested that mutations in GEMIN5 disrupt distinctive developmental and neurological pathways with slight overlap with SMA.

Since the mutations in GEMIN5 lead to a decrease in the snRNP assembly, we investigated the global splicing defects in GEMIN5[H913R] homozygous neurons compared to controls. We performed differential splicing analysis based on isoform expression by adjusting the threshold value to 5% and found 99 differentially spliced genes (DSGs) with a total of 440 isoforms in GEMIN5[H913R] compared to controls (Supplementary Fig. 13a). Functional enrichment analysis of the DSGs with FDR adjusted to <0.05 showed that overall ~93% of the genes undergo alternative splicing in GEMIN5[H913R] compared to controls (Supplementary Fig. 13b). This suggests that the variants of GEMIN5 disrupts snRNP assembly formation and might result in global splicing defects in the patient neurons.

## Discussion

Pathogenic variants in GEMIN5 have never been reported in the literature as a cause of human disease. We identified biallelic variants in GEMIN5 that give rise to a neurological syndrome which features developmental delay, cerebellar atrophy, and predominant motor dysfunction along with hypotonia.

Two of our families presented with severe symptoms in infancy with an SMA-like clinical picture combined with cerebellar hypoplasia, reminiscent of pontocerebellar hypoplasia type 1[55–57]. Most others presented with a childhood onset phenotype with a predominant cerebellar syndrome as well as ataxia, tremor, and hypotonia. In the latter group, hypotonia, motor developmental delay, and evidence of motor neuron disease on EMG in some patients draws further clinical similarities to an SMA-like motor neuronopathy (Supplementary Table 2). A small subset of individuals had slow onset progressive cerebellar symptoms along with appendicular spasticity reminiscent of spastic ataxia syndromes. All patients were observed to have some degree of cerebellar atrophy on MRI imaging (Fig. 1d–j, Supplementary Table 1, and clinical summary). The neonatal onset of symptoms in two families and non-progressive MRI findings in some of the patients with early childhood onset symptoms makes a case for cerebellar hypoplasia rather than atrophy in these cases. On the other hand, a subset of patients did have a progressive clinical phenotype, and some have had worsening of the cerebellar atrophy on imaging, suggesting a potential progressive nature of the cerebellar involvement in some patients.

Spinal muscular atrophies (SMAs) are a genetically and clinically heterogeneous group of conditions characterized by degeneration and loss of anterior horn cells in the spinal cord that lead to muscle weakness and atrophy[23,58,59]. Pontocerebellar hypoplasia type 1 (PCH1) is a condition characterized by pontocerebellar hypoplasia plus degeneration of motor neurons in the anterior horn of the spinal cord[55–57]. Many autosomal recessive genes including VRK1, TSEN54, ESOSC8, EXOSC3, EXOSC9,

TOE1, etc., have been implicated in this group[56,60,61]. Loss-of-function mutations in TOE1, a protein that encodes for deadenylase, have been identified in PCH7 patients and these mutations drastically reduce the expression of TOE1 protein in patient fibroblasts[61]. Mutating endogenous toe1 in zebrafish caused PCH-like defects including midbrain and hindbrain degeneration in vivo. Further mechanistic studies revealed that mutant TOE1 specifically associates with incompletely processed pre-snRNAs in PCH7 patient fibroblast cells[61]. Similarly, loss-of-function variants in the Integrator complex subunit 1 (INTS1) have been reported and linked with developmental delays, cataracts, and craniofacial anomalies[62]. Interestingly, loss of ints1 in a zebrafish model showed eye defects, similar to human patients, suggesting the role of the ints1 gene in eye development. Furthermore, loss of ints1 in zebrafish led to a reduction in proteins involved in the INT complex[62]. Also, disruption of the mouse U2 snRNA gene (NMF291−/−) has been shown to cause ataxia and neurodegeneration by perturbing global pre-mRNA splicing in a dosage-dependent manner[63]. The unique combination of a motor neuronopathy with cerebellar atrophy makes it difficult to classify these patients as SMA (due to presence of cerebellar involvement), PCH (due to presence of motor neuron involvements), or any other disease category. Given the unique spectrum of clinical presentation and GEMIN5 variants, we suggest classifying them currently as a distinct syndrome of GEMIN5 spectrum disease. We think it would be worth considering testing for GEMIN5 variants in SMN1 negative neonates with severe hypotonia and absent reflexes, especially with cerebellar atrophy on imaging. We also think GEMIN5 should be covered by ataxia gene panels and should be considered in children with motor predominant developmental delay and cerebellar atrophy on neuroimaging.

All variants were located in conserved alpha helixes of the GEMIN5 protein, and six highly conserved amino acid residues (p. Ser73Pro, p.His923Pro, p.Ser1000Pro, p.Leu1068Pro, p. His1364Pro, and p.Leu1367Pro) were replaced with a proline, which is well-known for disrupting alpha helix secondary structure causing premature bending of the peptide chain[64]. The GEMIN5 p.Asp704Glu was located next to Phe705, which is known to interact directly with small nuclear RNAs (snRNAs); therefore, this variant probably alters snRNA recognition function[19].

The majority of GEMIN5 variants appear to cause loss of function by reducing protein expression (Figs. 2 and 3) potentially by either destabilization, increased turnover, affecting adjacent protein residues, or through any other mechanism. It is possible that the broad clinical spectrum and variable disease course across patients could be caused by the difference in decreased levels of endogenous GEMIN5 protein. We observed a significant reduction of GEMIN5 protein in the cytoplasm in patient iPSC neurons, suggesting that reducing the endogenous levels might be deleterious to the neuronal function (Fig. 2). It is possible that the adverse effects observed in patients is due to loss of cytoplasmic function, independent of nuclear function, as nuclear GEMIN5 protein levels are unaffected (Fig. 2b and d). Since variants in GEMIN5 reduce the protein expression, it may have adverse effects on differential expression of RNA and protein targets. Most of the GEMIN5 variants were clustered in the linker-dimerization domain that connects the WD with the RBS domains which provides a platform for protein–protein/RNA interactions and dimerization[16,26,65]. It is possible that these functions are perturbed due to variants in GEMIN5 as evident from a significant loss of snRNP complex proteins in human patient-derived iPSC neurons (Figs. 2 and 3).

RNA-binding domains present in any RBPs exert multiple cellular functions such as RNA binding specificity, affinity, and

translation[66,67]. Apart from binding snRNPs, GEMIN5 has been shown to regulate global translation via the WD domains-mediated interaction with 60S ribosomal subunit, as well as selective translation through non-canonical RNA-binding sites (RBS1 and RBS2)[15,28,65,68]. The C-terminal part of GEMIN5 protein binds to a hairpin flanked by A/U/C-rich sequences in internal ribosome entry site (IRES) elements and regulates translational activity[28]. The presence of variants in the RBS domains as well as the spacer region raises the possibility of structural destabilization of the hairpins which in turn might cause translational dysregulation and reduced IRES binding. It is likely that mutant GEMIN5 protein might become unstable due to improper folding and ubiquitylation which targets it for degradation by proteasome or autophagy. We found accumulation of ubiquitin-positive puncta in iPSC neurons expressing GEMIN5 p.His913Arg variant suggesting that the protein degradation machinery might become activated (Supplementary Fig. 7).

We observed that GEMIN2 protein expression levels are also reduced along with GEMIN5 in patient neurons as well as in cells with shRNA-mediated GEMIN5 KD compared to controls (Figs. 3 and 4). Besides SMN and GEMIN5, GEMIN2 is an essential core component required for the assembly of the SMN complex. It binds to SMN and Sm heptameric rings to facilitate their interaction with GEMIN5-snRNA[69,70]. It has been reported that SMN–GEMIN2 interaction is abolished due to loss of function mutations of SMN1 protein in SMA patients. Furthermore, mouse studies have shown that reduced levels of GEMIN2 disrupt U snRNP complex formation leading to motor neuron degeneration[71]. This suggest that both GEMIN5 and GEMIN2 may have complementary functions.

Our data suggests GEMIN5 variants lead to loss-of-function of SMN complex assembly proteins (Figs. 2 and 3). The degree of endogenous GEMIN5 KD in mammalian cells correlates with the reduced expression of snRNP proteins, suggesting that over 50% loss of GEMIN5 protein might be required for causing any obvious symptoms (Fig. 4). This is important since haploinsufficiency does not seem to cause disease in humans, as all heterozygous carriers are asymptomatic. We are unable to rule out the possibility that loss of GEMIN5 protein might upregulate proteins which in turn lead to deleterious effects due to gain of function mechanism. Rigor mortis, Drosophila homolog of human GEMIN5, is highly expressed in the brain and known to regulate snRNP assembly and other functions similar to human protein[72,73]. Rigor mortis mutants show defects in molting, duplicated mouth parts and defects in puparium formation. Conditional ubiquitous RNA-mediated knock down of endogenous rigor mortis in Drosophila caused severe developmental defects, premature lethality, motor dysfunction, and reduced life span (Fig. 4j–p), similar to our patients with GEMIN5 variants showing motor predominant developmental delays.

GEMIN5 is involved in the assembly of the SMN protein complex via directly binding with SMN-snRNA-and the Sm protein core. Disruption in snRNP assembly has been shown to cause motor neuron degeneration in animal models and has been linked with SMA pathogenesis[18,19,25,71,74]. Using an in vitro reconstitution approach, we found that variants in GEMIN5 reduce snRNP assembly formation in iPSC neurons as well as in shRNA-mediated KD of GEMIN5 (Fig. 5a, b and d). The possible cause of reduced snRNP assembly could be loss of GEMIN5 protein levels as well as its disrupted interaction with other GEM proteins. By immunoprecipitation, we found that both L1068P and H913R mutations in GEMIN5 greatly reduced its interaction with GEMIN4 and GEMIN3, the proteins required to transfer the GEMIN5–pre-snRNA to the SMN–Gemin2–Sm protein complex (Fig. 5c). A previous study has shown that the human U1-specific

RBP, U1-70K can bridge pre-U1 to SMN–Gemin2–Sm, in a Gemin5-independent manner suggesting an alternative pathway for snRNP assembly[75]. The difference in the clinical presentations of SMA and GEMIN5 syndrome could be explained by the presence of non-canonical GEMIN5-independent snRNP complex formation in our patient neurons.

The RNA targets of GEMIN5 are largely unknown, hence we decided to perform RNA-sequencing to identify how variants in GEMIN5 alter RNA transcripts at a global level using patient-derived iPSC neurons (Fig. 6a and Supplementary Fig. 10a). Interestingly, most of the transcripts we identified were associated with neuronal development, translation, protein turn-over, and cellular signaling, further explaining that clinical features observed in our patients might be due to alteration in these physiological pathways.

GEMIN5 is an indispensable component of the SMN assembly complex and disruption of SMN assembly has been found in SMA, a lethal motor neuron degenerative disease caused by the loss of SMN1 protein[74]. With cerebellar hypotonia as one of the hallmarks and distinct features found in GEMIN5 patients, few of the patients shared clinical symptoms similar to SMA. Given the clinical and mutational heterogeneity among our GEMIN5 patients, it is challenging to accurately predict the clinical course as no genotype–phenotype correlation studies have been yet performed.

To explore if the manifestation of discrete but overlapping clinical symptoms in GEMIN5 patients is due to variability in genes and pathways, we compared the RNA-sequencing data between GEMIN5 (GEMIN5$^{H913R}$) and SMA (SMN1$^{Exon7del}$) patients. Surprisingly, the majority of the transcripts and pathways which are differentially regulated in GEMIN5$^{H913R}$ are unique and are not found in SMN1$^{Exon7del}$ (Fig. 6f). They are mostly involved in regulation of development processes, post-synaptic membrane organization, transport, and signal transmission. Interestingly, we found that SMN1$^{Exon7del}$ and GEMIN5$^{H913R}$ shared very few pathways which were involved in the autonomous nervous system, cell cycle arrest, and response to developmental stimuli (Fig. 6e). Even among the commonly shared transcripts between SMN1$^{Exon7del}$ and GEMIN5$^{H913R}$, most of them showed a differential and contrasting expression trend (Fig. 6a and d). Thus, the transcriptomic comparison between GEMIN5$^{H913R}$ and SMN1$^{Exon7del}$ revealed that although being a crucial part of the same snRNP assembly complex, mutations in GEMIN5 lead to an exclusive transcriptomic profile with little overlap to SMA (Fig. 6). The distinct but overlapping predisposition of clinical symptoms in GEMIN5 patients could be attributed to functions besides snRNP biogenesis and splicing, and needs further exploration for targeted therapeutic approaches.

In summary, we have shown that biallelic variants in GEMIN5 cause developmental delay, motor dysfunction, and cerebellar atrophy and reduce snRNP complex assembly proteins, impair snRNP assembly and misregulate RNA targets.

## Methods

**Exome sequencing.** Families 1, 11–13, and 15–18 were sequenced at GeneDx (Gaithersburg, MD). Using genomic DNA from the proband as well as parents and siblings, when available, the exonic regions and flanking splice junctions of the genome were captured using the SureSelect Human All Exon V4 (50 Mb), the Clinical Research Exome kit (Agilent Technologies, Santa Clara, CA) or the IDT x Gen Exome Research Panel v1.0. Massively parallel (NextGen) sequencing was done on an Illumina system with 100 bp or greater paired end reads. Reads were aligned to human genome build GRCh37/UCSC hg19 and analyzed for sequence variants using a custom-developed analysis tool. Exception is the patient from family 13 whose sequencing was done using the Ataxia Xpanded panel and lacked full WES analysis. Additional sequencing technology and variant interpretation protocol used were similar as ref. [29]. For WES analysis of family 3, In solution exome capture was performed using the SeqCap EZ Human Exome Kit v3.0 (Roche

Nimblegen, USA) with 100-bp paired-end read sequences generated on a HiSeq2000 (Illumina, Inc., USA) in the Centro Nacional de Análisis Genómico in Barcelona (CNAG). Single variants and insertions/deletions (indels) were identified using the GATK's best practices for germline SNP and Indel discovery in WES and annotated by the Annovar software. Copy number variation (CNV) was analyzed by R package Exome Depth.

The general assertion criteria for variant classification are publicly available on the GeneDx ClinVar submission page (http://www.ncbi.nlm.nih.gov/clinvar/submitters/26957/). We found a subset of our GEMIN5 patients through GeneMatcher (https://genematcher.org/statistics)[30,31]. All the variants are annotated by using the GEMIN5 NP_056280.2 reference transcript in GnomAD and the other databases to estimate the allelic frequency. The damaging index of GEMIN5 variants was determined by using various in silico prediction tools such as Polyphen2[32], Provean[33], SNAP2[34], MUpro[35], PhD SNP[36], and SIFT[37].

Genetic testing in all centers was performed either in the setting of routine diagnostic testing without the requirement for institutional ethics approval or within research settings approved by the ethical review boards of the respective institutions. All patient information has been deidentified. Informed consent was obtained from patients for publication at each site per local institution requirements by the authors.

**CRISPR/Cas9-mediated generation of IPSCs**

*Plasmid generation.* iPSC lines were generated by CRISPR/Cas9 technique[38]. Following sgRNA identification for the site of interest using the CRISPOR design tool[39], we cloned the sgRNA sequences into the pLentiCRISPR-V2 plasmid from the laboratory of Feng Zhang (AddGene #52961) following the protocol provided with the plasmid[40,41].

*Electroporation, selection, and growth of edited iPSCs.* Human ESCs or iPSCs were cultured in hPSC medium on mouse embryonic fibroblast (MEF) feeder cells with Rho Kinase (ROCK)-inhibitor (1.0 µM, Calbiochem, H-1152P) for 24 h prior to electroporation[41]. Cells were digested by TrypLE express Enzyme (Life Technologies) for 3–4 min, washed two times with DMEM/F12, and harvested in hPSC medium with 1.0 µM ROCK-inhibitor. Cells were dispersed into single cells, and $1 \times 10^7$ cells were electroporated with appropriate combination of plasmids in 500 µl of electroporation buffer (KCl 5 mM, MgCl$_2$ 5 mM, HEPES 15 mM, Na$_2$HPO$_4$ 102.94 mM, NaH$_2$PO$_4$ 47.06 mM, pH = 7.2) using the Gene Pulser Xcell System (Bio-Rad) at 250 V, 500 µF in 0.4 cm cuvettes (Phenix Research Products). Cells were electroporated in a cocktail of 15 µg of the pLentiCRISPRV2-Gemin5 sg1fwd plasmid and 100 µL of a 10 µM ssODN targeting the Gemin5 locus. This ssODN was non-complementary to the sgRNA sequence and consisted of 141 nucleotides —70 nucleotides upstream and 70 nucleotides downstream of the targeted base pair[42]. Following electroporation, cells were plated on MEF feeders in 1.0 µM ROCK inhibitor. At 24- and 72-h post-electroporation, cells were treated with puromycin (0.33 µg/ml, Invivogen, ant-pr-1) to select for cells containing the pLentiCRISPRV1-Gemin5 sg1fwd plasmid. Concurrent with puromycin treatment, the cells were fed with MEF-conditioned hPSC media containing 1.0 µM ROCK inhibitor. After removal of the puromycin at 96 h, cells were cultured in MEF-conditioned hPSC media until colonies were visible.

*Genotyping.* Single-cell colonies were manually selected and mechanically disaggregated. Genomic DNA was isolated from a portion of these colonies using QuickExtract DNA Extraction Solution 1.0 (Epicentre). Genotyping primers were designed flanking the mutation site, allowing amplification of this region using Q5 polymerase-based PCR (NEB). PCR products were identified via agarose gel and purified using a Zymoclean Gel DNA Recovery Kit (Zymo Research). Clones were submitted to Quintara Biosciences for Sanger sequencing to identify clones with the proper genetic modification.

*Off-target analysis.* To identify whether the CRISPR-Cas9 system produced any non-specific genome editing, we analyzed suspected off-target sites for genome modification. Using the five highest-likelihood off-target sites predicted by the CRISPOR algorithms[43], we designed genotyping primers to amplify these regions via Q5-polymerase PCR. PCR products were identified via agarose gel, purified using a Zymoclean Gel DNA Recovery Kit, and submitted to Quintara Biosciences for Sanger sequencing.

**Generation of induced pluripotent stem cells (iPSCs) from peripheral blood.**
PBMCs were isolated from whole blood processed and reprogrammed into iPSCs by the Stem Cell Core Facility at Northwestern to generate clonal iPSC lines from patient blood. All samples were banked and then processed together to minimize variability due to batch effects. When a low number of PBMCs were isolated from limited patient samples, erythroid cells were expanded using SFEM II media supplemented with cytokines SCF, IL-3, and EPO for subsequent iPSC reprogramming. When expanded to a sufficient number, a non-integrating Sendai viral-based reprogramming kit (CytoTune 2.0 from ThermoFisher) was used to introduce the four "Yamanaka reprogramming factors", OCT4, SOX2, KLF4, and MYC. Reprogrammed iPSCs were expanded on plates coated with hESC-qualified matrigel (Corning) and grown in mTeSR plus (Stem Cell Technologies). Clonal

iPSC-like colonies were selected, expanded, and characterized to pass several quality control standards. At least three colonies for each line were selected after meeting our criteria for morphology, growth, sterility, and iPSC marker expression. Cells were expanded and analyzed to ensure >80% of colonies are free of differentiated cells and readily expand following passaging. Routine testing was performed on each clonal line to ensure they were free of mycoplasma contamination; karyotype analysis was performed to ensure cells were free of abnormalities, and STR analysis was performed to validate the identity of the cells.

**Cell culture and differentiation of iPSCs into neuronal cells.** The iPSCs were differentiated into neuronal cells as described below[44]. The iPSCs were cultured and maintained in mTeSR™ 1 media (STEMCELL technologies) on Matrigel-coated plates. For differentiation, ~0.6 million cells were plated and let to grow for up to 80–90% confluency in mTeSR™ 1 for 2 days. For the first phase of differentiation, the confluent iPSC cells were grown for 6 days in N2B27 Neurobasal/DMEM-F12 medium (1:1 v/v) containing 1% N2 (Gibco, 17502–048), 2% B27 (Gibco, 17054–044), 1% Glutamax (Gibco), and non-essential amino acids (NEAA) (Gibco, 11140050) along with 10 µM SB431542 (STEMCELL technologies), 0.1 µM LDN (Sigma SML0559), 1 µM retinoic acid (RA) (Sigma R2625), 1 µM smoothened agonist (SAG, Cayman chemicals 11914). For day 7–14, cells were grown in N2B27 media supplemented with 1 µM RA, 1 µM SAG, 10 µM DAPT (Cayman, 13197),16 µM SU5406 (Cayman, 131825). On day 14, cells were dissociated using TrypLE/DNase I (Invitrogen) and cultured on poly-ornithine and laminin-coated coverslips or plates in neuronal media containing neurobasal medium, N2, B27, 0.4 mg/ml ascorbic acid (Sigma, A4403), 10 µg/ml human brain-derived neurotrophic factor (BDNF) (Peprotech, 45002), 10 µg/ml glial cell-derived neurotrophic factor (GDNF) (Peprotech, 45010), 10 µg/ml ciliary neurotrophic factor (CNTF) (Peprotech, 45013), 1% Glutamax, and NEAA. The cells were differentiated into neurons for 28 days and processed for subsequent IF and WB analysis.

**Immunofluorescence.** For IF, the neurons were fixed in 4% paraformaldehyde (PFA) for 10 min and blocked in 0.1% Triton-X in PBS and 5% normal goat serum for 10 min. The cells were treated overnight with the following antibodies: mouse anti-GEMIN5 (Millipore Sigma HPA037393, 1:1,000), mouse anti-GEMIN2 [2E17] (abcam ab6084, 1:500), mouse anti-GEMIN6/SIP2, (abcam ab88290, 1:500) rabbit anti-GEMIN4 (NOVUS Biologicals NB110-40591, 1:500), mouse anti-GEMIN3, clone 12H12 (Millipore Sigma 05-1533, 1:500), mouse anti-SMN (BD transduction 610646, 1:1000), rabbit anti-U1A (NOVUS Biologicals NBP2-53095, 1:2000), chicken anti-beta-III Tubulin (NOVUS Biologicals NB100-1612, 1:1000), goat anti-MAP2 (Synaptic System-188 004, 1:1000), and mouse anti-Ubiquitin. Alexa fluor-488, Alexa fluor-568, and Alexa fluor-647 secondary antibodies were used from Invitrogen. The cells were mounted using fluoroshield™ with DAPI (Sigma) and images were taken at 60× using Nikon A1-T216.3 confocal microscope.

**WB analysis.** Differentiated neurons were dissociated in TrypLE/DNase and cells were pelleted down at 250×g at room temperature. The cells were washed with PBS and lysed in RIPA buffer containing 150 mM NaCl, 50 mM NaF, 2 mM EDTA, 0.2 mM Na orthovanadate, 1% sodium deoxycholate, 2 mM DTT, 1% NP40, 0.1% SDS, and protease inhibitor (Roche 11836170001). The lysates were sonicated and centrifuged at 10,000×g for 15 min at 4 °C. The concentration of proteins in the supernatant were measured by Pierce™ BCA protein assay kit (Thermo Scientific 23227). Equal concentration of supernatant was boiled with 1× Laemmli buffer and the proteins were separated using 4–12% NuPage bis–Tris gel (Novex/Life Technologies). Protein was transferred onto nitrocellulose (Invitrogen IB23001) using the iBlot2 (Life Technologies 13120134). The blots were blocked in 2.5% Quick-Blocker reagent (EMB Millipore WB57-175GM) and probed overnight with the following antibodies: mouse anti-tubulin (SIGMA, 1:10,000) anti-GEMIN5 (Gen-Tex GTX130498, 1:1000), mouse anti-GEMIN2 [2E17] (1:2000), mouse anti-GEMIN6/SIP2 (1:5000) rabbit anti-GEMIN4 (1:2000), mouse anti-GEMIN3, clone 12H12 (1:1000), mouse anti-SMN (1:5000), and rabbit anti-U1A (NOVUS Biologicals NBP2–53095, 1:2000).

For immunoprecipitation, lysates were prepared from HEK cells expressing HA-tagged GEMIN5, L1068P, and H913R in 10 mM Tris–HCl (pH 7.5), 100 mM Nacl, 2.5 mM MgCl$_2$, 0.1% NP40, 2 mM DTT, 2.5 mM sodium orthovanadate and 1× protease inhibitor cocktail (invitrogen). The lysates were incubated with anti-HA antibody overnight at 4 °C and the HA–protein complex was pulled down by incubating with Protein A Dynabeads (Invitrogen) for 3 h at 4 °C. The proteins were denatured and probed for anti-HA, anti-GEMIN4, anti-GEMIN3, and SMN.

Secondary antibodies used were anti-mouse DYLight 800 and anti-rabbit 680 (Invitrogen, 1:10,000). The blots were imaged using LI-COR imager (Odyssey CLx). All the blots were run in triplicates and the integrated band densities were calculated using image studio software (LI-COR).

**mRNA stability and Gene expression analysis.** RNA was isolated from iPSC-derived differentiated neurons by using the PureLink™ RNA mini kit (Invitrogen), following the manufacturer's instructions. Around 500 ng of RNA was used to synthesize cDNA with oligodT by using iScript™ Reverse Transcription kit (BioRad). Quantitative PCR was performed in 20 µl reaction in 7300 Real Time PCR machine (Applied Biosystems) using custom design 5′ 6-FAM/ZEN/3′ IBFQ

IDT PrimeTime Assay set (Supplementary Table 5). Gene expression levels ($C_t$ values) were normalized with GAPDH as internal control. For qPCR validation in flies, RNA was isolated from three whole flies expressing the RNAi by using TRizol and gene expression was normalized with Tubulin. mRNA decay was designed as mentioned above by using relative transcript abundance after 0, 1, 2, 4, 6, and 8 h of ActD (Sigma A1410) treatment.

**In vitro snRNP assembly assay**. Cytoplasmic extracts from the differentiated neurons were prepared using NE-PER nuclear and cytoplasmic extraction kit (Thermo Scientific 78835) and the protein concentration were measured by Pierce™ BCA protein assay kit. U1snRNA were transcribed from gel-eluted and linearized DNA template by in vitro transcription using T7 RNA polymerase and m7G cap analog. pCp-Cy3 (Cytidine-5′-phosphate-3′-(6-aminohexyl) phosphate) (Jena Bioscience) was transferred to the 3′-hydroxyl group on U1snRNA by T4 RNA ligase (Thermo Fisher). The snRNP assembly reaction was carried out by incubating 5 µg of pCp-Cy3-labeled U1snRNAs with 50 µg of cytoplasmic extract, 10 µM tRNA, and 2.5 mM ATP at 30 °C for one and half hours[9,45]. The reaction mix were loaded onto native 6% TBE polyacrylamide gel (Novex/Life Technologies). The gel was run at 150 V at 4 °C and was imaged using LI-COR imager.

**RNA sequencing**. RNA was isolated from iPSC-derived differentiated neurons with homozygous and heterozygous GEMIN5 His913Arg variants by using the PureLink™ RNA mini kit (Invitrogen). RNA sequencing was performed using the BGISEQ-500 platform combining the DNA nanoball-based nanoarrays and step-wise sequencing using Combinational Probe Anchor Synthesis Sequencing Method. Reads were mapped to human reference genome (hg19) using Bowtie2, and gene expression level were calculated with RSEM. Between the samples Pearson correlation was calculated using cor and the differentially expressed genes with the fold change ≥ 2.00 adjusted $p$ value ≤ 0.05 were selected. The DEGs with a false discovery rate (FDR) of not larger than 0.01 were used for GO functional enrichment using phyper. Statistical analysis was performed by using R. Differential splicing detection was done using the NBSplice package in Bioconductor/R. The expression matrix at the transcript/isoform level generated using RSEM was used as the input[46,47]. Negative binomial generalized linear models were fitted at the gene level and allow the estimation of significant differences in isoforms relative expression values between the biological conditions. The significance threshold is set at 5%. The Database for Annotation, Visualization and Integrated Discovery (DAVID) v6.8 was used to functionally annotate the DSGs into different pathways[48,49]. The Upkeyword pathways were plotted for genes with FDR < 0.05.

For comparing the SMA-sequencing data with GEMIN5 (His913Arg), both datasets were processed in a similar way. The SMA (SMN1Exon7del) RNA-sequencing data was obtained from Answer ALS, a large-scale resource for sporadic and familial ALS combining clinical data with multi-omics data from induced pluripotent cell lines (https://www.answerals.org). Quality controlled FASTQ files were aligned to the Ensemble Human reference genome (hg38) using STAR aligner (version 2.5.1). HTSeq-count were used to generate counts of reads uniquely mapped to annotated genes using the GRCh38 annotation gtf file[50]. Differential gene expression analysis between the different conditions was done using DESeq2[51] using a model based on the negative binomial distribution. The resulting $p$-values were adjusted using the Benjamini and Hochberg's approach for controlling the false discovery rate, and differentially expressed genes were determined at the 5% threshold. Gene set enrichment analysis was used to assess the statistical enrichment of gene ontologies, and pathways[52].

**Larval eclosion assay**. UAS-rigor mortis KK RNAi lines (VDRC 105403) were crossed with inducible driver Tubulin-GS-Gal4, at 28 °C on food mixed with 1 mM RU486 (Cayman Chemicals) for inducing transgene expression. The larvae were monitored from the 1st instar stage until they eclosed and become adults. The images of each developmental stage were taken using a Leica M205C dissection microscope equipped with a Leica DFC450 camera.

**Motor dysfunction assays**. UAS-rigor mortis KK RNAi lines (VDRC 105403) were crossed with ubiquitous inducible driver, Tubulin Gene switch (TubGS)-Gal4. Day 1 adults from the F1 progeny were collected every 24 h and moved to standard media mixed with 20 mM RU486 at 28 °C. Locomotion was assessed using the RING assay[53,54]. Briefly, flies were transferred, without anesthetization, into plastic vials and placed in the RING apparatus. The vials were tapped down against the bench and the climbing was recorded on video at day 20. Quantifications were performed manually by a third party in a blinded manner.

For studying NMJs defects, 3rd instar larvae expressing rig mortis RNAi were dissected, and fixed by using 4% formaldehyde. The RNAi was expressed using TubGS-Gal4 by growing the 1st instar larvae on 1 mM RU486 at 28 °C until they reach the 3rd instar stage. The larvae ($n = 4$) were probed with mouse anti-horseradish peroxidase (HRP), a presynaptic neuronal marker to identify the NMJs, for overnight at 4 °C. On the next day, the larvae were washed with 0.1% TBST and stained with goat anti-mouse Alexa fluor-568 secondary antibody. The larvae were mounted with fluroshield™ (Sigma) and images were taken at ×60 using Nikon A1-T216.3 confocal microscope.

**Life span assay**. Lifespan assay was performed on day 1 adult females. Female flies expressing the transgene for rig mortis RNAi by using TubGS-Gal4 were separated and transferred on to experimental vials containing fly food mixed with RU486 (20 mM) at a density of 25 flies per vial ($n > 100$). Deaths were scored every other day and flies were transferred to fresh food three times a week.

**Statistical analysis**. Statistical analysis was done on GraphPad Prism using one-way analysis of variance (ANOVA) followed by a Bonferroni or Tukey post hoc test for comparison between two or more groups. To compare two experimental conditions, two-tailed non-parametric Mann–Whitney $U$ test was performed. For analysis of mRNA stability, normalized values for 0, 1, 2, 4, 6, and 8 h were fitted to the non-linear regression of one phase-exponential decay model and half-lives were calculated using the equation, $t_{1/2} = \ln 2/k$.

**Reporting summary**. Further information on research design is available in the Nature Research Reporting Summary linked to this article.

## Data availability
RNA-sequencing data that support the findings of this study are available in the Gene Expression Omnibus (GEO) database under accession number GSE168622. Source data are provided with this paper.

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

## Acknowledgements

This work was supported by the Children's Neuroscience Institute Research grant (D.S.R. and U.B.P.), NIH grants R01NS073873 (J.E.L.), R01NS098004 (J.G.G.). Sequencing was provided by The Yale Center for Mendelian Genomics (UM1HG006504) and supported by NHGRI. This study was supported by the URDCat program (PERIS SLT002/16/00174), the Hesperia

Foundation and the Secretariat for Universities and Research of the Ministry of Business and Knowledge of the Government of Catalonia [2017SGR1206] to A.P. We are indebted to Juan José Martínez for technical expertise. The work in C.G.B. lab is supported by intramural funds from the NINDS, sequencing and analysis were provided by the Broad Institute of MIT and Harvard Center for Mendelian Genomics (Broad CMG) and was funded by the National Human Genome Research Institute, the National Eye Institute, the National Heart, Lung and Blood Institute, and grant UM1 HG008900 to Daniel MacArthur and Heidi Rehm. This study was supported in part by a core grant to the Waisman Center from the National Institute of Child Health and Human Development (U54 HD090256) and by a UW2020 Grant awarded to Anita Bhattacharyya and Su-Chun Zhang by the University of Wisconsin and the Wisconsin Alumni Research Foundation. We are thankful to Dr. Andrew Petersen, Dr. Randel Tibbetts, and Dr. Angel Alvarez for their help in generating the iPSC lines. This work was supported by the Stem Cell Core Facility at Northwestern University Feinberg School of Medicine. The computational analysis was performed using the high performance cluster hosted by the Center for Research Computing, University of Pittsburgh. We are thankful to Dr. Livio Pellizzoni for sharing the U1 snRNA construct.

## Author contributions

Design of this study: S.K., D.S.R., T.R.F., U.B.P.; Acquiring clinical/genetic info and data analysis: E.N.A., C.W., Y.L., S.L., Y.B.S., J-H.C., M.C., K.S., V.C., J.A., E.S.S., S.S.B., M.A. C., D.C., K.M., B.K., A.H.N., M.I., D.L., P.F., A.C., M.G., M.Y., I.D., N.B., C.G., S.L.R., K. K.S., R.U., A.G-O., A.N.O., E.V., A.P., H.R.M., J.E.L., S.A., E.C.A., S.B.M., C.P., M.J.G.S., A.B., H.H., J.K., D.S., S.R-S., T.M.S., S. Leiz, K.J., R.R., Y.Y., Y.Z., M.W., J.W., X.W., K.P., S.D., C.G.B., M. Wagner, M.Y.I., H.M.E., V.S., R.M., J.G.G., M.S.Z., J.S. Recruitment and management of study: S.K., D.S.R., T.R.F., U.B.P.; Bioinformatics and statistical analysis: D.R., S.K., T.R.F.; Wrote the manuscript: S.K., D.S.R., T.R.F., D.R., U.B.P.; Critically read and edited the manuscript: S.K., D.S.R., T.R.F., E.S.S., C.A.M., D.C., K.M., B.K., P.F., M. Y., A.P., J.E.L., E.C.A., S.B.M., M.J.G.S., A.B., D.S., S.D., C.G.B., R.M., M.S.Z., J.S., U.B.P. All authors reviewed and approved the manuscript prior to submission.

## Competing interests

J.E.L. is a member of the scientific advisory board for Cerevel Therapeutics. J.E.L. is a consultant and may provide expert testimony for Perkins Coie LLP. All other authors declare no competing interests.

## Additional information

[1]Department of Pediatrics, Childrens Hospital of Pittsburgh, University of Pittsburgh Medical Center, Pittsburgh, PA, USA. [2]Department of Biomedical Sciences, Seoul National University College of Medicine, Seoul, Republic of Korea. [3]Department of Rehabilitative Medicine, Pusan National University School of Medicine, Pusan, Republic of Korea. [4]Department of Pediatrics, Seoul National University College of Medicine, Seoul, Republic of Korea. [5]Developmental Brain Disorders Laboratory, Paris University, Imagine Institute, INSERM UMR, Paris, France. [6]Department of Genetics, AP-HP, Necker Enfants Malades Hospital, Paris University, Imagine Institute, Paris, France. [7]Greenwood Genetic Center, Greenwood, SC, USA. [8]Department of Laboratory Medicine and Pathology, Mayo Clinic, Rochester, MN, USA. [9]Center for Individualized Medicine, Mayo Clinic, Rochester, MN, USA. [10]Department of Pediatrics and Neurology, University of Texas Southwestern Medical Center, Dallas, TX, USA. [11]University of Mississippi Medical Center, Jackson, MS, USA. [12]Division of Genetics, University of Mississippi Medical Center, Jackson, MS, USA. [13]Nuffield Department of Clinical Neurosciences, University of Oxford, Oxford, UK. [14]Oxford Centre for Genomic Medicine, Oxford University Hospitals National Health Service Foundation Trust, Oxford, UK. [15]Department of Pediatrics, Division of Health Informatics, Childrens Hospital of Pittsburgh, Pittsburgh, PA, USA. [16]Department of Medical Genetics and Alberta Children's Hospital Research Institute, Cumming School of Medicine, University of Calgary, Calgary, Canada. [17]Department of Biochemistry and Medical Genetics, Rady Faculty of Health Sciences, University of Manitoba, Winnipeg, MB, Canada. [18]Department of Pediatrics and Neurology, Children's Hospital of Colorado, University of Colorado School of Medicine, Aurora, CO, USA. [19]Department of Pediatric Neurology, AP-HP, Necker Enfants Malades Hospital, Paris University Imagine Institute, Paris, France. [20]Department of Pediatric Radiology, AP-HP, Necker Enfants Malades Hospital, Paris University Imagine Institute, Paris, France. [21]Department of Pediatric Neurophysiology AP-HP, Necker Enfants Malades Hospital, Paris University, Paris, France. [22]Department of Neurology, Oslo University Hospital, Oslo, Norway. [23]Department of Research and Development, Division of Neuroscience, Oslo University Hospital and the University of Oslo, Oslo, Norway. [24]Department of Clinical Biochemistry, Institut de Recerca Sant Joan de Déu and CIBERER, Barcelona, Spain. [25]Hospital Universitario Miguel Servet, Zaragoza, Spain. [26]Hospital Sant Joan de Déu, Barcelona, Spain. [27]Centre for Biomedical Research on Rare Diseases (CIBERER), Instituto de Salud Carlos III, Madrid, Spain. [28]Catalan Institution for Research and Advanced Studies (ICREA), Barcelona, Spain. [29]Center for Mendelian Genomics, Broad Institute of MIT and Harvard, Cambridge, MA, USA. [30]Department of Neurology, University of Massachusetts Medical School, Worcester, MA, USA. [31]Department of Neurological Surgery, University of Pittsburgh School of Medicine, Pittsburgh, PA, USA. [32]Department of Pediatrics, University of Texas Health Science Center, Houston, TX, USA. [33]Department of Pediatrics, Division of Human Genetics, Rhode Island Hospital and Warren Alpert Medical School of Brown University, Providence, RI, USA. [34]GeneDx, Gaithersburg, MD, USA. [35]Department of Neuromuscular Diseases, UCL Queen Square Institute of Neurology, London, UK. [36]Department of Neuropediatrics and Muscle Disorders, Medical Center,, Faculty of Medicine, University of Freiburg, Freiburg, Germany. [37]Division of Human Genetics, Medical University Innsbruck, Innsbruck, Austria. [38]Institute of Human Genetics, Faculty of Medicine, Technical University Munich, Munich, Germany. [39]Clinic for Children and Adolescents Dritter Orden, Divison of Neuropediatrics, Munchen, Germany. [40]Department of Neurology, Gillette Children's Specialty Healthcare, St Paul, MN, USA. [41]Department of Pediatrics, Peking University First Hospital, Beijing, China. [42]The First People's Hospital of Changde City, Hunan, China. [43]Cipher Gene Ltd, Beijing, China. [44]Institute of Human Genetics, University of Leipzig Medical Center, Leipzig, Germany. [45]Neuromuscular and Neurogenetic Disorders of Childhood Section,

National Institute of Neurological Disorders and Stroke, National Institutes of Health, Bethesda, MD, USA. [46]Institute of Human Genetics, Klinikum rechts der IsarTechnical, University of Munich, Munich, Germany. [47]Clinical Genetics Department, Human Genetics and Genome Research Division, National Research Centre, Cairo, Egypt. [48]Departments of Neurosciences and Pediatrics, Rady Children's Institute for Genomic Medicine, Howard Hughes Medical Institute, University of California, San Diego, La Jolla, CA, USA. [49]Department of Neurology, Friedrich-Baur-Institute, University Hospital, LMU Munich, Munich, Germany. [50]Department of Human Genetics, Graduate School of Public Health, University of Pittsburgh, Pittsburgh, PA, USA. [51]Children's Neuroscience Institute, Children's Hospital of Pittsburgh, University of Pittsburgh Medical Center, Pittsburgh, PA, USA. [52]These authors contributed equally: Sukhleen Kour, Deepa S. Rajan. ✉email: udai@pitt.edu

