## [Peer Review File · Nature Communications]

REVIEWER COMMENTS

Reviewer #1 (Remarks to the Author):

In this manuscript, Kour et al reported novel biallelic variants in the GEMIN5 gene from dozens of patients with developmental delay, hypotonia, and cerebellar ataxia. Using patient and CRISPR/Cas9-engineered iPSC differentiated neurons, they further examined and found that two of the disease-causing mutants (L1068P and H913R) exhibit a disturbed subcellular expression and localization of GEMIN5 and GEMIN2, but not SMN, GEMIN4 or GEMIN6. They also observed significantly decreased levels of GEMIN proteins, SMN and U1A in the two mutants. Similar pathological features were found in the fly strain with RNAi knockdown of dm-GEMIN5. Finally, they found distinct and unique transcriptomic signature between the iPSC differentiated neurons of GEMIN5 and SMA patients.

Overall, this is a well-organized manuscript with significant study that identifies many pathogenic variants in GEMIN5, and their effects to localization and stability of related proteins. This work is also important for understanding many neural related diseases. However, several concerns should be addressed.

Major:

1. Twenty-six variants of GEMIN5 are reported, the authors did not provide selection reason for the two variants (Leu1068Pro and His913Arg). Are they typical?
2. Similarly, for all the variants they identified, without further evidence to show they are the cause, calling "Disease-causing GEMIN5 variants" are too strong.
3. Lethality and climbing ability are not sufficient to argue that the RNAi fly has neuron defects. Other kind of assay should be provided.
4. GEMIN proteins and SMN are responsible for biogenesis of snRNPs, they are not components of snRNP complex. All the related descriptions should be clarified.
5. As it is critical for the assembly of snRNPs, transcriptomic analysis should not be limited in the gene expression level. Obviously, splicing and alternative splicing should be analyzed and discussed.

Minor:

1. U1 snRNP has a protein component U1A, all the places with "UA1" protein in this manuscript, I guess, should be U1A.
2. In the figures with western blotting, signals of tubulin are too strong. Overexposed or saturated signals are not proper for a loading control.

I would be interested in a revised version that accounted for these concerns.

Reviewer #2 (Remarks to the Author):

Overall, the authors show convincing data that GEMIN5 mutations are involved in a complex symptom complex of developmental delay and cerebellar signs. They show evidence in iPSC cells of a distinct set of transcripts being changed, changes in various proteins of the protein complex including GEMIN5, and show also motor function defects in a fly model. Overall this is of interest to a large audience so my comments for revision are relatively minor

1) There are numerous English grammar / structure problems in the manuscript - too many for me to list all - examples are lines 135, 201, 202 but there are many more. Needs to be remedied.

I found the abstract misleading:

2) "pathogenic variants in GEMIN5 ... result in a distinct neurological cerebellar ataxia syndrome." This is not correct. The combination of developmental delay, hypotonia and cerebellar ataxia with MRI-confirmed hypoplasia is common to dozens of disorders and not distinct. Moreover, the phenotypic spectrum of the patients is from infantile onset with death before age 5 to adult onset, so extremely wide. This phenotypic spectrum is recognized in the main part of the paper and the heterogeneous, not specific nor distinct spectrum of severity needs to be acknowledged in the abstract as well.

3) "we observed that GEMIN5 variants disrupt distinct, yet overlapping, set of transcripts and pathways as compared to SMA patient neurons" - when later it becomes clear that the overlap is very minor, not significant and often in a different direction than SMA. It would be better to state just distinct from SMA.

4) Fig 2A - move Sanger sequencing to supplementary materials.

Reviewer #3 (Remarks to the Author):

This is a comprehensive and elaborate work, providing robust genetic and functional evidence for this novel gene GEMIN5 in association with an autosomal recessive, early-onset cerebellar ataxia and hypotonia syndrome.

The gene GEMIN5 is a small nuclear ribonuclear protein (snRNP) involved in the formation of spliceosomes and thereby responsible for the regulation of other target proteins. The authors identified 28 individuals from 20 families carrying 26 different, biallelic, loss-of-function and missense variants in GEMIN5, associated with hypotonia, motor development delay, cerebellar atrophy, and ataxia. Using patient-derived induced pluripotent stem cells (iPSC), Kour et al. were able to show a reduced overall expression and disturbed cytoplasmic distribution of GEMIN5 in motor neurons. This was hypothesized to be related to mRNA and protein instability and not to transcriptional misregulation (cycloheximide chase and western blot, ubiquitination patterns, and actinomycin D assay). In HEK cells, the authors showed by small hairpin RNA knockdown that GEMIN5 levels have a dosage-dependent effect on SMN expression. Using immunoprecipitation studies, they explained the disruption of snRNP assembly by demonstrating a disturbed interaction of GEMIN5 with SMN, GEMIN4, and GEMIN3. siRNA-based downregulation of the GEMIN5-homologue in a drosophila model resulted in developmental disturbances, motor dysfunction (rapid iterative negative geotaxis), and a reduced life-span. In iPSC-derived neurons, RNA-seq analyses

revealed differences between biallelic GEMIN5 mutation carriers and patients with SMN-related proximal spinal muscle atrophy (SMA), which was interpreted as a potential explanation for phenotypic differences between SMA and GEMIN5-associated cerebellar ataxia. Overall, this original paper of high interest and strong genetic support. Language and style could be improved; some examples of many more are given below. Several questions regarding methods and functional results remain:

1. Be consistent with Oxford comma throughout the manuscript
2. Figure 2g: “zoom”?? Do you mean magnification?
3. GEMIN5 levels in cells: not sure how this quantification is normalized across plates to avoid batch effects and normal variance in staining intensity
4. Figure 5a and b: rather than intensity alone, wouldn't you expect a shift in size in a faulty assembly? What is the positive control for this experiment?
5. p. 4, l. 149f: “the effects of disrupting snRNP complex 150 dynamics in the pathogenesis of other disorders has not been studied”: please compare pathomechanism and clinical phenotypes in Lardelli et al., 2017 (TOE1) and Krall et al., 2019 (INTS1). In Jia et al., 2011, NMF291 deficient mice had higher rates of abnormal splicing in cerebellum and developed truncal ataxia.
6. p. 5, l. 166: Disease-causing GEMIN5 mutations (??) cause
7. p. 5, l. 171f: “an ataxia multi-gene panel which included trinucleotide repeat analysis”: PLs check if this is correct for NGS panels
8. Patient 1: were the parents consanguineous? What is the allele frequency of the variant in healthy reference populations?
9. p. 6, l. 181: The term „developmental concerns” seems somewhat inappropriate here. Do you mean delay? Please specify.
10. P. 6, l. 188f: “Central hypotonia was a common clinical feature in most patients”: Please be more concrete here: How many patients had hypotonia, how many had spasticity, and was there a clinical overlap? How did you determine central from peripheral hypotonia? Was any of the patients examined by nerve conduction studies or EMG to assess the hypothesis of motor neuron disease or concomitant neuropathy/myopathy?
11. P. 6, l. 189: “All ambulatory patients appeared to be ataxic”: A pure gait ataxia? How did you determine that it was of cerebellar and not of afferent origin?
12. P. 6, l. 190f: “Some of the patients were felt to have a static phenotype”: This is not a matter of feeling. Were symptoms progressive or not? Please give concrete numbers or percentages (e.g.: X patients showed a progressive, X patients a stable course).
13. P. 6, l. 193-200: Language: “were noted”, “were felt”. Better “we observed cerebellar atrophy in...”, “X patients had”, “brain MRI revealed...”. Please do not forget the comma before the “and” in enumerations.
14. P. 7, l. 202f: “All variants involved residues that are evolutionary conserved across different species and are rare or absent in gnomAD.” Please mention the range in allele frequency.
15. Please mention that (if?) compound-heterozygosity was confirmed in all non-homozygous cases.
16. P. 12, l. 353: “downregulated genes in GEMIN5 compared to SMA”: GEMIN5 is a gene, SMA a disease (that is mostly, but not always associated with the SMN gene). Please specify.
17. P. 12, ll. 353ff: “By comparing the significant DEGs in SMA and GEMIN5 as shown by Venn diagram, we identified the genes which are either common between GEMIN5 and SMA or specific to GEMIN5 only. We identified 1278 and 3004 transcripts unique to GEMIN5 and SMA, respectively, whereas 622 transcripts are shared 357 among these two disease conditions.” Some duplication here, please shorten and re-phrase.
18. P. 13, l. 371: “involved in the development of the autonomic nervous system”

19. P. 13, l. 373: "However, the majority of pathways..."
20. P. 13, l. 373f: "which explains why 374 GEMIN5 and SMA patients show different clinical presentations": This line of argument is somewhat unsupported and not based on functional evidence yet. I would suggest to formulate it a little more conservatively like "might explain".
21. P. 29, l. 757: How can developmental delay be of adult onset?
22. P. 29, l. 762: "hypotonia, motor developmental delay, and evidence of motor neuron disease on EMG". You haven't mentioned these EMG changes before. They should be briefly characterized in the results part as well.
23. P. 31, l. 797f: Is it possible to proof this by your results? E.g. by correlating expression levels with a disease severity score or with the age of onset?
24. P. 32, l. 820: "suggesting that the protein degradation machinery might become..."
25. P. 32, l. 827: In SMA, the mutations are not primarily in the SMN protein, but the SMN protein is not expressed due to deletions of the gene.
26. Figure 1A: Pedigrees: Please explain what the little bumps between the parents mean. Consanguinity? Traditionally, the arrow indicates the family proband. As all affected family members apparently have been examined, I would suggest to just mention this in the legends and reserve the arrow for its usual purpose. Does the plus mean wild-type? Why don't you just say wt instead, to make it clearer?
27. Figure 1C: Schematic: Why did you show the homozygous variants only? Aren't the compound-heterozygous variants interesting as well?
28. Figure 1e, h, i: On the presented MRI sections, it seems like your patients had not only cerebellar, but pontocerebellar atrophy?
29. Figure legends: Please explain the abbreviations somewhere in the legends. Markers such as TUJ1 or MAP2 should be explained.
30. P. 16, l. 435 and p. 17, l. 47: The Student's t test is a parametric test. A non-parametric variant would be the Kruskal-Wallis test. Did you use an independent or dependent t-test in Figure 3m, n, and o? Please explain. Which method did you use for alpha-error correction (post-test analysis) following ANOVA?
31. P. 16, l. 446: the part "eclosion defects" is not correctly labeled with (i). Figure 4(l) is not well explained.

Response to the reviewer's comments:

We are very grateful to the reviewers for their thoughtful comments and suggestions that have tremendously helped us improve the quality of our current manuscript. All three reviewers acknowledged the significance of our work revealing the identification of novel variants in GEMIN5. We strongly believe that we have appropriately addressed the issues raised by the reviewers. We are providing a point-to-point response to the comments below.

Reviewer 1: Reviewer #1 (Remarks to the Author):

In this manuscript, Kour et al reported novel biallelic variants in the GEMIN5 gene from dozens of patients with developmental delay, hypotonia, and cerebellar ataxia. Using patient and CRISPR/Cas9-engineered iPSC differentiated neurons, they further examined and found that two of the disease-causing mutants (L1068P and H913R) exhibit a disturbed subcellular expression and localization of GEMIN5 and GEMIN2, but not SMN, GEMIN4 or GEMIN6. They also observed significantly decreased levels of GEMIN proteins, SMN and U1A in the two mutants. Similar pathological features were found in the fly strain with RNAi knockdown of dm-GEMIN5. Finally, they found distinct and unique transcriptomic signature between the iPSC differentiated neurons of GEMIN5 and SMA patients.

Overall, this is a well-organized manuscript with significant study that identifies many pathogenic variants in GEMIN5, and their effects to localization and stability of related proteins. This work is also important for understanding many neural related diseases. However, several concerns should be addressed.

Response: We thank the reviewer for commenting on the importance and significance of our current work. We appreciate your kind words on the organization of our manuscript.

Major:

Comment 1: Twenty-six variants of GEMIN5 are reported, the authors did not provide selection reason for the two variants (Leu1068Pro and His913Arg). Are they typical?

Response: The reviewer raised a valid point. The Leu1068Pro was the first GEMIN5 variant identified in our clinic about 4 years ago and we had the blood samples of the patient as well as an unaffected parent available for reprogramming them into iPSC lines. Our Leu1068Pro patient was in the mild to moderate spectrum of clinical symptoms. We came to know about the His913Arg family (consanguineous) in Germany where these patients were seen by Prof. Jan Senderak (a co-author in the manuscript). Two brothers harboring His913Pro variant (family 4) died within few months after birth and this is the strongest variant we found so far. Since these patients were deceased, we used CRISPR/cas9 approach to introduce this variant in a healthy control iPSC line and generated an isogenic control for our study. We selected the Leu1068Pro and His913Arg variants because these variants were identified during the initial phases of our

study. We have clearly listed the recurrent variants in the Supplementary Figure 2 and Supplementary table 3.

Comment 2. Similarly, for all the variants they identified, without further evidence to show they are the cause, calling “Disease-causing GEMIN5 variants” are too strong.

Response: We agree with the reviewer. In each of these patients, no other significant variant(s) came up in our exome analysis and allelic frequencies of these variants (homozygous) have been reported as zero in publicly and privately available databases (**Supplementary Table 2**). These variants are highly conserved across the species. Hence, we strongly believe that these variants are the likely cause of the symptoms observed in our patients. In the light of the reviewer’s comment, we are toning down our statement and listing our variants as likely associated with the disease.

Comment 3. Lethality and climbing ability are not sufficient to argue that the RNAi fly has neuron defects. Other kind of assay should be provided.

Response: To address the reviewer’s comment, we have performed neuromuscular junction (NMJ) analysis (added to **figure 4**) to examine the impact of knocking down endogenous rigor mortis (fly version of human GEMIN5) using a neuron-specific driver. The fruit flies (*Drosophila*) are a powerful genetic model system for examining neuromuscular development and function in different human neurological diseases. We found a significant reduction in the bouton size in the larvae with rigor mortis RNAi knockdown as compared to the controls. We are including this data in **figure 4m** and o. In summary, we provide strong and compelling evidences that reducing the endogenous levels of rigor mortis by RNAi leads to 1) premature lethality; 2) NMJ defects and motor dysfunctions; and 3) reduced life span in flies (**Figure 4 j-p**).

Comment 4. GEMIN proteins and SMN are responsible for biogenesis of snRNPs, they are not components of snRNP complex. All the related descriptions should be clarified.

Response: As the reviewer suggested, we have clarified the descriptions about snRNPs biogenesis throughout the text.

Comment 5. As it is critical for the assembly of snRNPs, transcriptomic analysis should not be limited in the gene expression level. Obviously, splicing and alternative splicing should be analyzed and discussed.

Response: This is the first report showing identification of novel variants in GEMIN5 among 30 affected individuals and we have done a significant amount of work towards examining the functional consequences of these variants in iPSC neurons as well as in flies. GEMIN5 is a multifunctional protein and it is important for regulating protein and RNA metabolism. We are in a process of doing in-depth analysis of splicing and alternative splicing events in patient neurons and examining how these aberrant splicing alterations are responsible for causing

clinical symptoms in our GEMIN5 patients. We plan to focus more on this as part of a larger follow up study.

We did perform *in silico* analysis of our RNA-sequencing data and specifically looked for any splicing alternations in the GEMIN5 patient iPSC neurons and are including that splicing information (**Supplemental Figure 13 and Supplementary Excel File**). We are providing a list of differentially spliced genes (DSGs) in the patient iPSC neurons expressing His913Arg variant in GEMIN5. In addition to this, we have done pathway analysis using DSGs and including that data as a supplementary figure as well.

Minor:

Comment 6. U1 snRNP has a protein component U1A, all the places with “UA1” protein in this manuscript, I guess, should be U1A.

Response: We have corrected this mistake in our revised manuscript.

Comment 7. In the figures with western blotting, signals of tubulin are too strong. Overexposed or saturated signals are not proper for a loading control.

Response: Thank you for this comment. We have provided better quality Western blots with low exposure of tubulin signals. These experiments were repeated 3-5 times and quantified.

Comment 8: I would be interested in a revised version that accounted for these concerns.

Response: We thank the reviewer for giving us an opportunity to revise our manuscript and address the concerns. We hope the reviewer will appreciate our efforts to address his/her concerns.

Reviewer #2 (Remarks to the Author):

Overall, the authors show convincing data that GEMIN5 mutations are involved in a complex symptom complex of developmental delay and cerebellar signs. They show evidence in iPSC cells of a distinct set of transcripts being changed, changes in various proteins of the protein complex including GEMIN5, and show also motor function defects in a fly model. Overall, this is of interest to a large audience so my comments for revision are relatively minor

Response: We greatly appreciate your positive feedback on our current work. We agree that our work would be of great interest to a large audience. We have fully addressed the comments raised by the reviewer.

Comment 1: There are numerous English grammar / structure problems in the manuscript - too many for me to list all - examples are lines 135, 201, 202 but there are many more. Needs to be remedied.

Response: Our manuscript has been read through thoroughly by multiple authors and all identified grammatical mistakes have been corrected. The paper revision has also been edited by a professional scientific editor who has further improved the grammar and structure of the manuscript. We thank the reviewer for pointing this out.

Comment 2: I found the abstract misleading: "pathogenic variants in GEMIN5 ... result in a distinct neurological cerebellar ataxia syndrome." This is not correct. The combination of developmental delay, hypotonia and cerebellar ataxia with MRI-confirmed hypoplasia is common to dozens of disorders and not distinct. Moreover, the phenotypic spectrum of the patients is from infantile onset with death before age 5 to adult onset, so extremely wide. This phenotypic spectrum is recognized in the main part of the paper and the heterogeneous, not specific nor distinct spectrum of severity needs to be acknowledged in the abstract as well.

Response: We thank the reviewer for picking up on this. We agree that the clinical phenotype of neurodevelopmental delay and ataxia is not distinct by itself. We have changed the abstract to indicate accordingly. In addition to this, we modified the abstract and the title to address this issue.

Comment 3: "we observed that GEMIN5 variants disrupt distinct, yet overlapping, set of transcripts and pathways as compared to SMA patient neurons" - when later it becomes clear that the overlap is very minor, not significant and often in a different direction than SMA. It would be better to state just distinct from SMA.

Response: We agree. We have changed the wording to "we observed that GEMIN5 variants disrupt a distinct set of transcripts and pathways as compared to SMA patient neurons".

Comment 4: Fig 2A - move Sanger sequencing to supplementary materials.

Response: As the reviewer suggested, we moved Sanger sequencing panel to supplementary materials (**Supplementary Fig 3a and b**).

Reviewer #3 (Remarks to the Author):

This is a comprehensive and elaborate work, providing robust genetic and functional evidence for this novel gene GEMIN5 in association with an autosomal recessive, early-onset cerebellar ataxia and hypotonia syndrome.

The gene GEMIN5 is a small nuclear ribonuclear protein (snRNP) involved in the formation of spliceosomes and thereby responsible for the regulation of other target proteins. The authors identified 28 individuals from 20 families carrying 26 different, biallelic, loss-of-function and missense variants in GEMIN5, associated with hypotonia, motor development delay, cerebellar atrophy, and ataxia. Using patient-derived induced pluripotent stem cells (iPSC), Kour et al. were able to show a reduced overall expression and disturbed cytoplasmic distribution of

GEMIN5 in motor neurons. This was hypothesized to be related to mRNA and protein instability and not to transcriptional misregulation (cycloheximide chase and western blot, ubiquitination patterns, and actinomycin D assay). In HEK cells, the authors showed by small hairpin RNA knockdown that GEMIN5 levels have a dosage-dependent effect on SMN expression. Using immunoprecipitation studies, they explained the disruption of snRNP assembly by demonstrating a disturbed interaction of GEMIN5 with SMN, GEMIN4, and

GEMIN3. siRNA-based downregulation of the GEMIN5-homologue in a drosophila model resulted in developmental disturbances, motor dysfunction (rapid iterative negative geotaxis), and a reduced life-span. In iPSC-derived neurons, RNA-seq analyses revealed differences between biallelic GEMIN5 mutation carriers and patients with SMN-related proximal spinal muscle atrophy (SMA), which was interpreted as a potential explanation for phenotypic differences between SMA and GEMIN5-associated cerebellar ataxia.

Overall, this original paper of high interest and strong genetic support. Language and style could be improved; some examples of many more are given below. Several questions regarding methods and functional results remain:

Response: We are grateful the reviewer for going through our paper in-depth and giving his/her constructive feedback that allowed us to improve the quality and presentation of work. We agree that our work would be of broad interest and is supported by strong genetic data.

Comment 1: Be consistent with Oxford comma throughout the manuscript

Response: We apologize for this mistake. We, as well as a professional scientific editor, have gone through the manuscript and made Oxford comma consistent.

Comment 2. Figure 2g: “zoom”?? Do you mean magnification?

Response: By Zoom, we meant enlarging one given inset (cell) from the image taken at 60X magnification. We have clarified this in the methods sections as well.

Comment 3: GEMIN5 levels in cells: not sure how this quantification is normalized across plates to avoid batch effects and normal variance in staining intensity

Response: To avoid batch effects, we compared the neurons from the same batch of differentiations and took all the images at 60X magnification. Please note that we have differentiated the neurons in different batches as well to rule out any batch-to-batch ambiguity and variation. All the steps of immunocytochemistry were kept consistent across the batch. Furthermore, we measured the integrated density value of each individual cell normalized to each cell soma unit area to prevent the variation between the cells across different images.

Comment 4: Figure 5a and b: rather than intensity alone, wouldn't you expect a shift in size in a faulty assembly? What is the positive control for this experiment?

Response: The binding assay we utilized to study the assembly formation *in vitro* used biotin labeled U1snRNA to detect the complex instead of using protein specific antibodies so depending upon the degree of interaction of GEMIN5 and other SMN complex proteins with U1snRNA, we see either complete or partial loss of assembly measured by intensity instead of shift. Since mutations in GEMIN5 resulted in decrease levels and interaction of GEMIN5 with other complex protein, we observed decrease in the assembly formation instead of complete loss. This assay has been successfully used to examine SMN complex formation by multiple labs and published (Pellizzoni *et al.*, 2002). In this study, we used HEK293 transfected with scrambled construct as a positive control for assembly formation *in vitro*.

Comment 5: p. 4, l. 149f: “the effects of disrupting snRNP complex 150 dynamics in the pathogenesis of other disorders has not been studied”: please compare pathomechanism and clinical phenotypes in Lardelli *et al.*, 2017 (TOE1) and Krall *et al.*, 2019 (INTS1). In Jia *et al.*, 2011, NMF291 deficient mice had higher rates of abnormal splicing in cerebellum and developed truncal ataxia.

Response: We thank the reviewer for pointing out these references. We compared the clinical phenotypes in Lardelli *et al.*, 2017 (TOE1) and Krall *et al.*, 2019 (INTS1) with our patients and we are including the following statements in the Discussion section.

“Pontocerebellar hypoplasias (PCH) are a heterozygous group of devastating conditions characterized by structural abnormalities in the brainstem, specifically the pons and the cerebellum. Many autosomal recessive genes including VRK1, TSEN54, ESOSC8, EXOSC3, EXOSC9, and TOE1 etc., have been implicated in this group. Loss-of-function mutations in TOE1, a protein that encodes for deadenylase, have been identified in PCH7 patients and these mutations drastically reduce the expression of TOE1 protein in patient fibroblasts. Mutating endogenous *toe1* in zebrafish caused PCH-like defects including midbrain and hindbrain degeneration *in vivo*. Further mechanistic studies revealed that mutant TOE1 specifically associates with incompletely processed pre-snRNAs in PCH7 patient fibroblast cells (Lardelli *et al.*, 2017).

Loss-of-function variants in the Integrator complex subunit 1 (INTS1) have been reported and linked with developmental delays, cataracts, and craniofacial anomalies (Krall *et al.*, 2019). Interestingly, loss of *ints1* in a zebrafish model showed eye defects, similar to human patients, suggesting the role of the *ints1* gene in eye development. Furthermore, loss of *ints1* in zebrafish led to a reduction in proteins involved in the INT complex (Krall *et al.*, 2019).

Disruption of the mouse U2 snRNA gene (NMF291 *-/-*) has been shown to cause ataxia and neurodegeneration by perturbing global pre-mRNA splicing in a dosage-dependent manner (Jia *et al.*, 2012).

Comment 6: p. 5, l. 166: Disease-causing GEMIN5 mutations (??) cause

Response: We changed this statement to “to novel biallelic variants in GEMIN5”. Correct

Comment 7: p. 5, l. 171f: “an ataxia multi-gene panel which included trinucleotide repeat analysis”: PLs check if this is correct for NGS panels

Response: The ataxia multi-gene panel was mentioned in specific for our index patient. For other patients, where the NGS platform was used, the platform by itself did not include triplet repeats. Most of these patients had separate triplet repeat testing to exclude these disorders.

Comment 8: Patient 1: were the parents consanguineous? What is the allele frequency of the variant in healthy reference populations?

Response: Excellent point! We also wondered about the possibility of consanguinity in the patient 1. Although the parents denied about being related to each other, their families come from a small geographical area. Interestingly, no suggestion of consanguinity was detected on the genetic data. The allelic frequency of Leu1068Pro in gnomAD database has been reported to 3.98e-5 as heterozygote; no homozygote has been ever reported (**Supplementary table 2**).

Comment 9: p. 6, l. 181: The term “developmental concerns” seems somewhat inappropriate here. Do you mean delay? Please specify.

Response: Thank you for this comment. We have changed this to developmental delays.

Comment 10: P. 6, l. 188f: “Central hypotonia was a common clinical feature in most patients”: Please be more concrete here: How many patients had hypotonia, how many had spasticity, and was there a clinical overlap? How did you determine central from peripheral hypotonia? Was any of the patients examined by nerve conduction studies or EMG to assess the hypothesis of motor neuron disease or concomitant neuropathy/myopathy?

Response: Thank you for this observation. We have made changes to the manuscript as suggested. Central and appendicular hypotonia was per the clinical neurologist assessment at each site. 23 out of the 30 patients were noted to have central hypotonia. We observed that 13 out of the 30 patients had concomitant appendicular hypertonia/ spasticity. 1 We also found that 6 out of the 30 patients had an EMG/ NCV completed with 10 of these suggestive of neuropathic or motor neuron disease. The details are attached in a separate table in the supplementary data, which is now included in the manuscript (**Supplementary table 2**).

Comment 11: P. 6, l. 189: “All ambulatory patients appeared to be ataxic”: A pure gait ataxia? How did you determine that it was of cerebellar and not of afferent origin?

Response: All ambulatory patients were noted to have a gait ataxia. At this time, it is unclear if the ataxia is purely cerebellar in nature and we believe there could be a component of the hypotonia and the motor neuron disease that could contribute to the gait ataxia in these

patients, which is consistent with the cerebellar atrophy. We have changed the manuscript to indicate that the ambulatory patients had a gait ataxia.

Comment 12: P. 6, l. 190f: "Some of the patients were felt to have a static phenotype": This is not a matter of feeling. Were symptoms progressive or not? Please give concrete numbers or percentages (e.g.: X patients showed a progressive, X patients a stable course).

Response: Thank you for this comment. We have modified this statement. We observed a static phenotype in 15 out of the patients and a progressive phenotype in 6 out of the 30 patients. Due to the ongoing pandemic situation, data on clinical follow up was unavailable at that time of the manuscript for 9 out of the 30 patients.

Comment 13: P. 6, l. 193-200: Language: "were noted", "were felt". Better "we observed cerebellar atrophy in...", "X patients had", "brain MRI revealed...". Please do not forget the comma before the "and" in enumerations.

Response: We have modified the language as the reviewer suggested.

Comment 14: P. 7, l. 202f: "All variants involved residues that are evolutionary conserved across different species and are rare or absent in gnomAD." Please mention the range in allele frequency.

Response: We have included a table showing allelic frequency of each variant identified in our current study. Please see Supplementary table 2.

Comment 15: Please mention that (if?) compound-heterozygosity was confirmed in all non-homozygous cases.

Response: Compound heterozygosity was confirmed by segregation with parents or derived from trio WES, in all non-homozygous cases and we are providing Sanger sequencing data in **Supplementary figure 1**.

Comment 16: P. 12, l. 353: "downregulated genes in GEMIN5 compared to SMA": GEMIN5 is a gene, SMA a disease (that is mostly, but not always associated with the SMN gene). Please specify.

Response: Excellent point and we agree with the reviewer and have changed the analysis to focus on the comparison of the individual mutations of each disease, GEMIN5 (**GEMIN5^{H913R}**) and SMA (**SMN1^{Ex7del}**), instead of referring to the two diseases to compare.

Comment 17: P. 12, ll. 353ff: "By comparing the significant DEGs in SMA and GEMIN5 as shown by Venn diagram, we identified the genes which are either common between GEMIN5 and SMA or specific to GEMIN5 only. We identified 1278 and 3004 transcripts unique to GEMIN5 and SMA, respectively, whereas 622 transcripts are shared 357 among these two

disease conditions.” Some duplication here, please shorten and re-phrase.

Response: We agree and have rephrased the text as reviewer suggested.

Comment 18: P. 13, l. 371: “involved in the development of the autonomic nervous system”

Response: We thank the reviewer for catching this typo. The term “Autonomous” was corrected to “autonomic nervous system”.

Comment 19: P. 13, l. 373: “However, the majority of pathways...”

Response: We made the necessary changes.

Comment 20: P. 13, l. 373f: “which explains why 374 GEMIN5 and SMA patients show different clinical presentations”: This line of argument is somewhat unsupported and not based on functional evidence yet. I would suggest to formulate it a little more conservatively like “might explain”.

Response: As the reviewer suggested, we formulated the statement and toned it down to “might explain”.

Comment 21: P. 29, l. 757: How can developmental delay be of adult onset?

Response: Thank you for pointing out this error. This statement has been modified accordingly.

Comment 22: P. 29, l. 762: “hypotonia, motor developmental delay, and evidence of motor neuron disease on EMG”. You haven’t mentioned these EMG changes before. They should be briefly characterized in the results part as well.

Response: Thank you for catching this. We have modified the results section now to include EMG data and have attached a table in the supplementary data with the EMG results in the 16 out of the 30 patients that had it (Supplementary table 4).

Comment 23: P. 31, l. 797f: Is it possible to proof this by your results? E.g. by correlating expression levels with a disease severity score or with the age of onset?

Response: Excellent point! We did try to prove our points by knocking down endogenous GEMIN5 protein in HEK293 cells (**Figure 4**) where we found that the degree of knockdown of GEMIN5 protein correlates with the degree of reduction in other GEMIN5 interacting proteins (GEMIN2, GEMIN3 and GEMIN4). We are in a process of generating additional iPSC lines from GEMIN5 patients carrying variants in different domains/parts of the protein. We plan to differentiate these iPSC lines into neurons and perform the experiment that the reviewer proposes. It would take us at least a year to get these things accomplished and is beyond the scope of this manuscript.

Comment 24: P. 32, l. 820: “suggesting that the protein degradation machinery might become...”

Response: We made the changes in the text.

Comment 25: P. 32, l. 827: In SMA, the mutations are not primarily in the SMN protein, but the SMN protein is not expressed due to deletions of the gene.

Response: We made the necessary changes in the text.

Comment 26: Figure 1A: Pedigrees: Please explain what the little bumps between the parents mean. Consanguinity? Traditionally, the arrow indicates the family proband. As all affected family members apparently have been examined, I would suggest to just mention this in the legends and reserve the arrow for its usual purpose. Does the plus mean wild-type? Why don't you just say wt instead, to make it clearer?

Response: We used the little bumps in the double lines to indicate consanguinity. Since the reviewer pointed it out, we removed the little bumps and used double lines to show consanguinity. We changed the nomenclature from plus sign to WT and clarified in the figure legends as reviewer suggested.

Comment 27: Figure 1C: Schematic: Why did you show the homozygous variants only? Aren't the compound-heterozygous variants interesting as well?

Response: We totally agree. If we include a schematic with homozygous variants and compound heterozygotes, the figure becomes too crowded making it difficult to draw any conclusions. To avoid this, we decided to make two figures, one with only homozygous variants matching the pedigrees in the **figure 1** and one with compound heterozygotes as a supplementary **figure 2**. We have done our functional characterization work only on homozygous variants, so we have shown them as a separate figure. We strongly believe that both homozygous and compound heterozygous variants are equally important.

Comment 28: Figure 1e, h, i: On the presented MRI sections, it seems like your patients had not only cerebellar, but pontocerebellar atrophy?

Response: Thank you for this astute pick up! We agree and do feel that there might be a component of pontocerebellar atrophy which is why we mention the spectrum of pontocerebellar hypoplasia in the discussion. However, we are still in the process of doing a detailed neuroradiological phenotyping of these patients to establish this with certainty.

Comment 29: Figure legends: Please explain the abbreviations somewhere in the legends. Markers such as TUJ1 or MAP2 should be explained.

Response: We have clarified the abbreviations in the legends.

Comment 30: P. 16, l. 435 and p. 17, l. 47: The Student's t test is a parametric test. A non-parametric variant would be the Kruskal-Wallis test. Did you use an independent or dependent t-test in Figure 3m, n, and o? Please explain. Which method did you use for alpha-error correction (post-test analysis) following ANOVA?

Response: We apologize for the mistake. We used parametric, unpaired Student's t test for comparison between two groups. We corrected the text in the legends as well as in method section. For the posttest multiple comparison, we applied Tukey test following ANOVA.

Comment 31: P. 16, l. 446: the part "eclosion defects" is not correctly labeled with (i). Figure 4(l) is not well explained.

Response: We apologize for this mistake. We have corrected it in **figure 4** and explained it properly. We are citing references where eclosion defects have been used as a readout.

REVIEWER COMMENTS

Reviewer #1 (Remarks to the Author):

I am satisfied with the authors' response and revise.

Reviewer #2 (Remarks to the Author):

My comments have been adequately addressed.

Given the clinical heterogeneity and the heterogeneity of static vs. progressive disease, which the authors discuss, authors should add a sentence in the discussion acknowledging that when any new GEMIN5 mutations are found (e.g. prenatally or at birth), the clinical course will be difficult to predict, as no strict genotype-phenotype correlation studies have been performed.

Reviewer #3 (Remarks to the Author):

The authors have answered the questions in sufficient detail. No further comments from this reviewer.

Response to the reviewers' comments:

We are grateful to the reviewers' for accepting our manuscript for publication. The reviewer 2 suggested to add a statement about heterogeneity among our patients and lack of genotype-phenotype correlation. We have fully addressed this comment below and also in the discussion section.

Reviewer #1 (Remarks to the Author):

Comment: I am satisfied with the authors' response and revise.

Response: Thanks for accepting our manuscript.

Reviewer #2 (Remarks to the Author):

Comment: My comments have been adequately addressed.

Given the clinical heterogeneity and the heterogeneity of static vs. progressive disease, which the authors discuss, authors should add a sentence in the discussion acknowledging that when any new GEMIN5 mutations are found (e.g. prenatally or at birth), the clinical course will be difficult to predict, as no strict genotype-phenotype correlation studies have been performed.

Response: We agree. The reviewer raised a very valid point. We have added the following statement in the discussion section.

“Given the clinical and mutational heterogeneity among our GEMIN5 patients, it is challenging to accurately predict the clinical course as no genotype-phenotype correlation studies have been yet performed”

Reviewer #3 (Remarks to the Author):

Comment: The authors have answered the questions in sufficient detail. No further comments from this reviewer.

Response: We thank the reviewer for accepting our paper.